# DenoiseVAE: Learning Molecule-Adaptive Noise Distributions for Denoising-based 3D Molecular Pre-training

**Yurou Liu**[1,†], **Jiahao Chen**[1,†], **Rui Jiao**[2,3,†], **Jiangmeng Li**[4], **Wenbing Huang**[1, *], **Bing Su**[1, *]
[1] Gaoling School of Artificial Intelligence, Renmin University of China, Beijing, China
[2] Department of Computer Science and Technology, Tsinghua University, Beijing, China
[3] Institute for AI Industry Research, Tsinghua University, Beijing, China
[4] Institute of Software, Chinese Academy of Sciences, Beijing, China
{yurouliu99, nicelemon666}@gmail.com, jiaor21@mails.tsinghua.edu.cn,
jiangmeng2019@iscas.ac.cn

## Abstract

Denoising learning of 3D molecules learns molecular representations by imposing noises into the equilibrium conformation and predicting the added noises to recover the equilibrium conformation, which essentially captures the information of molecular force fields. Due to the specificity of Potential Energy Surfaces, the probabilities of physically reasonable noises for each atom in different molecules are different. However, existing methods apply the shared heuristic hand-crafted noise sampling strategy to all molecules, resulting in inaccurate force field learning. In this paper, we propose a novel 3D molecular pre-training method, namely DenoiseVAE, which employs a Noise Generator to acquire atom-specific noise distributions for different molecules. It utilizes the stochastic reparameterization technique to sample noisy conformations from the generated distributions, which are inputted into a Denoising Module for denoising. The Noise Generator and the Denoising Module are jointly learned in a manner conforming with the paradigm of Variational Auto Encoder. Consequently, the sampled noisy conformations can be more diverse, adaptive, and informative, and thus DenoiseVAE can learn representations that better reveal the molecular force fields. Extensive experiments show that DenoiseVAE outperforms the current state-of-the-art methods on various molecular property prediction tasks, demonstrating the effectiveness of it.

## 1 Introduction

Molecular representation learning plays an important part in many areas, including material design (Bishara et al., 2023; Ha et al., 2023), life science (Eslami et al., 2022; Goshisht, 2024) and drug discovery (Blanco-Gonzalez et al., 2023; Mak et al., 2023). Due to the critical role of 3D structures in determining molecular properties and the severe insufficiency of labeled data, various studies have explored 3D molecular pre-training (Zhou et al., 2023; Luo et al., 2022; Liu et al., 2022a). Among them, denoising has been proven to be one of the most effective approaches (Ni et al., 2023; Liu et al., 2022a; Feng et al., 2023; Jiao et al., 2024), which adds noises to atomic coordinates of the equilibrium conformation and reconstruct the equilibrium conformation from the noisy structure. Since denoising is equivalent to learning the molecular force fields (Vincent, 2011), it can capture the dynamic properties of molecules, which is beneficial for various downstream tasks.

Previous denoising methods mainly focus on exploring the noise-adding strategies for sampling noisy conformations of molecules. 3D-EMGP (Jiao et al., 2023) and Coordinate denoising (Coord) (Zaidi et al., 2022) both treat the 3D conformation of the molecule as a random quantity sampled from the Boltzmann distribution and apply invariant-scale Gaussian noises to all atoms among different molecules. They ignore that the molecular force fields are anisotropic (Feng et al., 2023;

---

*Correspondence to: Bing Su <subingats@gmail.com>, Wenbing Huang <hwenbing@126.com>.
† indicates equal contribution.

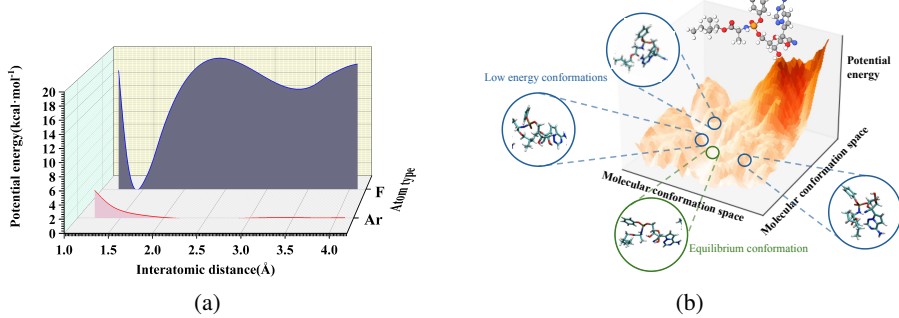

Figure 1: **(a)** Variation curves of potential energy with interatomic distance for different types of atoms. When the interatomic distance changes the same, the energy changes of the two atoms are different, which illustrates the anisotropy of the force fields. **(b)** The Potential Energy Surface (PES) of Remdesivir. Sampling from positions near the equilibrium conformation on the PES can yield conformations with different structures but also low energies, which is our learning objective.

Ni et al., 2023), which can be demonstrated by Fig. 1(a), i.e., the reasonable position changes of different atoms in physics are different. To tackle this issue, Frad (Feng et al., 2023) proposes a hybrid noise generation strategy that combines dihedral angle noise and coordinate noise. SliDe (Ni et al., 2023) develops a more fine-grained strategy on this basis. Although these hand-crafted strategies allow for the application of different noises to different atoms, they are designed by expertise heuristics and treat all molecules equally.

As shown in Fig. 1(b), molecules are dynamically variant, continuously moving within 3D Euclidean space and creating a Potential Energy Surface (PES). The minima on the PES corresponds to the equilibrium conformation of the molecule, indicating that the molecular system has reached equilibrium. However, beyond the equilibrium state, the low energy conformations in the figure are all observed when the molecular system is in equilibrium. In essence, the continuous changes in conformation are the equilibrium state of this molecular system. Therefore, generating physically reasonable noises to sample diverse and meaningful conformations around the minima of the PES is crucial for the denoising task to learn accurate force fields (Schlegel, 2003; Liu et al., 2022a; Zheng et al., 2024). The PESs for different molecules are highly specific and the feasible variation ranges for different atomic positions in different molecules may vary greatly (Sato, 1955). For example, atoms connected by single bonds may have a larger range of variation, while if the position of atoms in conserved functional groups undergoes significant changes, the properties of the molecule may change, or even the molecule may become another molecule; the variation range of the same atom or functional group varies in different molecule environments. It is unreasonable to share the same noise design among different molecules, as in existing state-of-the-art methods including Frad (Feng et al., 2023) and SliDe (Ni et al., 2023). Moreover, hand-crafted strategies are difficult to adapt to these complex and ever-changing situations. Therefore, we locate the dilemma that sharing the same hand-crafted noise generation strategies for all molecules leads to inaccurate learning of molecule-specific force fields.

To simultaneously take into account the anisotropy of the force fields within the molecule and the PES differences among different molecules, we propose a new pre-training method, named DenoiseVAE, that generates specific noise distributions for each atom within different molecules to adaptively sample molecular conformations with a Noise Generator. The adaptive noises and the randomness brought by the reparameterization process largely expand the sampling space of molecular conformations. The sampled conformations are fed into a Denoising Module for denoising. DenoiseVAE jointly trains the Noise Generator and the Denoising Module in a Variational Auto Encoder (VAE) (Kingma & Welling, 2013) manner due to its superior representational capacity and the solid theoretical guarantee. Through better aligning with physical principles, DenoiseVAE outperforms existing denoising methods in a variety of downstream tasks. Our main contributions are summarized as follows:

- We propose a denoising-based 3D molecular pre-training method, i.e., DenoiseVAE, which employs a learnable noise generating strategy instead of existing hand-crafted strategies to adaptively acquiring atom-specific noise distributions for different molecules.

- We propose a variational method to jointly optimize the Denoising Module and the Noise Generator of DenoiseVAE, where the denoising objective encourages the generated noises to be more consistent with the physical constraints so that equilibrium conformations can be recovered, while a KL divergence-based regularization term is imposed to prevent low noise intensity and increase the diversity of sampled conformations.

- Theoretically, we prove that optimizing our pre-training objective is equivalent to improving the evidence lower bound (ELBO) of the log-likelihood.

- We conduct extensive experiments to demonstrate the effectiveness of DenoiseVAE. Results show that DenoiseVAE outperforms existing denoising methods on various datasets for both molecular and complex property prediction.

## 2 RELATED WORK

**Molecular pre-training via denoising**  Pre-training is an important method for molecular representation learning. Inspired by the field of computer vision (Vincent et al., 2008), denoising as a self-supervised learning task is widely used in 3D molecular pre-training and achieves excellent results in many downstream tasks (Jiao et al., 2023; Zhou et al., 2023; Ni et al., 2023). Current denoising methods mainly differ in the added noise distribution and denoising tasks. Uni-Mol (Zhou et al., 2023) combines the coordinate denoising task with the atom-level masking strategy. They add uniform noise within a fixed scale to the atom coordinates and reconstruct the original coordinates and the atom-pair distances. Transformer-M (Luo et al., 2022) performs multi-view learning of molecules and jointly trains 2D graph structure and 3D coordinates. Coord (Zaidi et al., 2022) focuses more on the denoising task itself. They add Gaussian noise to the equilibrium conformation and predict the noise to learn the molecular force fields. Considering that the molecular energy should be invariant to rotation and translation, GeoSSL-DDM (Liu et al., 2022a) utilizes Gaussian noise to denoise the distance between atomic pairs. 3D-EMGP (Jiao et al., 2023) uses fixed multi-scale Riemann-Gaussian noise to perform coordinate denoising tasks. To satisfy the anisotropy of the force fields in different parts of the molecule, Frad (Feng et al., 2023) proposes a hybrid noise that combines dihedral angle noise and coordinate noise but only denoises the coordinate part. SliDe (Ni et al., 2023) optimizes the noise strategy on this basis, which achieves better conformation sampling by perturbing bond lengths, angles, and torsion angles. Compared with previous methods, our work is an attempt at how to sample more significant and various molecular conformations.

**Molecular property prediction with 3D information**  Understanding 3D molecular structures and properties is crucial for material design and drug discovery (Han et al., 2024). While some studies use 3D data during training, they rely solely on 2D representations for inference, limiting their ability to learn 3D conformations. (Stärk et al., 2022; Zhu et al., 2022; Liu et al., 2021). In order to fully utilize the 3D information of molecules, Transformer-M (Luo et al., 2022) performs multi-view learning by jointly training the 2D graph and 3D structure of the molecule, which explicitly uses the 3D coordinates of the molecule as input. Based on this, various subsequent works further use 3D DFT conformations to train the model for more accurate 3D molecular property prediction (Jiao et al., 2023; Zaidi et al., 2022; Zhou et al., 2023; Lu et al., 2023; Feng et al., 2023; Ni et al., 2023; Jiao et al., 2024). Graphormer-3D (Shi et al., 2022) utilizes the initial 3D conformations provided by the OC20 dataset (Chanussot et al., 2021) to predict the equilibrium energy. 3D-EMGP (Jiao et al., 2023) and Coord (Zaidi et al., 2022) design pre-training tasks for approximate molecular force field learning, and perform molecular property prediction on the molecular dynamics dataset (Ramakrishnan et al., 2014). Considering the anisotropy of the molecular force fields, Frad (Feng et al., 2023) and SliDe (Ni et al., 2023) improve the noise design in the denoising tasks, which are more accurately aligned with physical principles. Models such as EGHN (Han et al., 2022), FastEGNN (Zhang et al., 2024), and EquiRNA (Li et al., 2025a) better model molecules through hierarchical or virtual node approaches. On the other hand, spherical-scalarization models (*e.g.* SO3KRATES (Frank et al., 2024), HEGNN (Cen et al., 2024), GotenNet (Aykent & Xia, 2025)) untilize invariant information (inner product or modulus) from high-degree representations to enhance the expressive power of models. More recently, EquiLLM (Li et al., 2025b) further enhances the representation effect through knowledge injection from large language models. EPT (Jiao et al., 2024) conducts cross-domain joint training in the hope of learning more complementary knowledge across domains to optimize the results of molecular property prediction.

## 3 METHOD

### 3.1 PRELIMINARY

According to the prior knowledge in the field of statistical physics, the probability of the occurrence of 3D molecular conformation is described by Boltzmann distribution (Boltzmann, 1868):

$$p_{\text{physical}}(\boldsymbol{X}) = \frac{1}{Z} \exp(-\frac{E(\boldsymbol{X})}{kT}), \tag{1}$$

where $\boldsymbol{X}$ denotes the 3D conformation randomly sampled from the Boltzmann distribution and $E(\boldsymbol{X})$ represents the potential energy of the given molecule. $k$, $T$, and $Z$ denote the Boltzmann constant, temperature coefficient, and normalized factor, respectively.

Due to the challenge and high cost of measuring the true force fields within molecules, it is appealing to learn the force field for each atom within a molecule, represented as $-\nabla_{\boldsymbol{X}} E(\boldsymbol{X})$. By taking the logarithm of Eq. 1 and computing the gradient with respect to $x$, we obtain $\nabla_{\boldsymbol{X}} \log p(\boldsymbol{X}) = -\nabla_{\boldsymbol{X}} E(\boldsymbol{X})$, where $-\nabla_{\boldsymbol{X}} \log p(\boldsymbol{X})$ is the score function of conformation $\boldsymbol{X}$. From previous studies (Vincent, 2011), we typically solve for $p(\tilde{\boldsymbol{X}}|\boldsymbol{X})$ to approximate the true $p(\boldsymbol{X})$, since the potential energy distribution $E(\boldsymbol{X})$ and the true conformation distribution $p_{\text{physical}}(\boldsymbol{X})$ of molecules are unobtainable. Here, $\tilde{\boldsymbol{X}}$ represents molecular conformations sampled after noise perturbation. Previous studies (Vincent, 2011; Zaidi et al., 2022; Feng et al., 2023) have demonstrated the equivalence between the denoising task and learning force fields, specifically:

$$
\begin{aligned}
&\mathbb{E}_{p(\tilde{\boldsymbol{X}})}\big\|\text{GNN}_\Psi(\tilde{\boldsymbol{X}}) - \big(-\nabla_{\tilde{\boldsymbol{X}}} E(\tilde{\boldsymbol{X}})\big)\big\|^2 \\
=&\mathbb{E}_{p(\tilde{\boldsymbol{X}})}\big\|\text{GNN}_\Psi(\tilde{\boldsymbol{X}}) - \nabla_{\tilde{\boldsymbol{X}}} \log p(\tilde{\boldsymbol{X}})\big\|^2 \\
\propto&\mathbb{E}_{p(\tilde{\boldsymbol{X}},\boldsymbol{X})}\big\|\text{GNN}_\Psi(\tilde{\boldsymbol{X}}) - \nabla_{\tilde{\boldsymbol{X}}} \log p(\tilde{\boldsymbol{X}} \mid \boldsymbol{X})\big\|^2 \\
=&\mathbb{E}_{p(\tilde{\boldsymbol{X}},\boldsymbol{X})}\big\|\text{GNN}_\Psi(\tilde{\boldsymbol{X}}) - \frac{(\boldsymbol{X} - \tilde{\boldsymbol{X}})}{\boldsymbol{\sigma}^2}\big\|^2,
\end{aligned}
\tag{2}
$$

where $\text{GNN}_\Psi(\tilde{\boldsymbol{X}})$ denotes the atom-level noise predictions using a graph neural network (GNN) parameterized by $\Psi$, which takes the noisy conformation $\tilde{\boldsymbol{X}}$ as input.

Therefore, 3D molecular representation learning adopting the denoising task as the training objective can learn molecular force fields. Formally, given a set of unlabelled 3D molecular conformations $S = \{(\boldsymbol{X}_i)\}_{i=1}^{M}$, where $M$ is the size of the set, the objective of denoising is to train a Denoising Module, which is constituted by an encoder and a prediction head, to predict the added noises for the input noisy conformation, so that the acquired representations can capture the dynamic information of the potential force fields. As shown in Eq.3, the optimal parameters $\Psi^*$ of the Denoising Module can be obtained by:

$$\boldsymbol{\Psi}^* = \arg\min_{\boldsymbol{\Psi}} \frac{1}{M} \sum_{i=1}^{M} \big\|\text{GNN}_{\boldsymbol{\Psi}}(\tilde{\boldsymbol{X}}_i) - \frac{(\boldsymbol{X}_i - \tilde{\boldsymbol{X}}_i)}{\boldsymbol{\sigma}^2}\big\|^2. \tag{3}$$

From Eq. 2 we can see that a better noise generation strategy for conformation sampling is the key to more accurate learning of molecular force fields.

### 3.2 DENOISEVAE

Previous research in computational chemistry (Schlegel, 2003) has shown that molecules are not static but exist in a continuous state within 3D Euclidean space. These states form a PES, as illustrated in Fig.1(b). Since conformations corresponding to lower potential energy are more stable and have a higher existence in the real world, sampling more conformations around the minima of a molecule's PES is crucial for learning better molecular representations.

Previous methods (Feng et al., 2023; Ni et al., 2023) has considered the anisotropy of force fields within a molecule and added mixed noises by perturbing dihedral angle and bond angle, etc. However, such complex hand-crafted strategies ignore that different molecules should not share the same

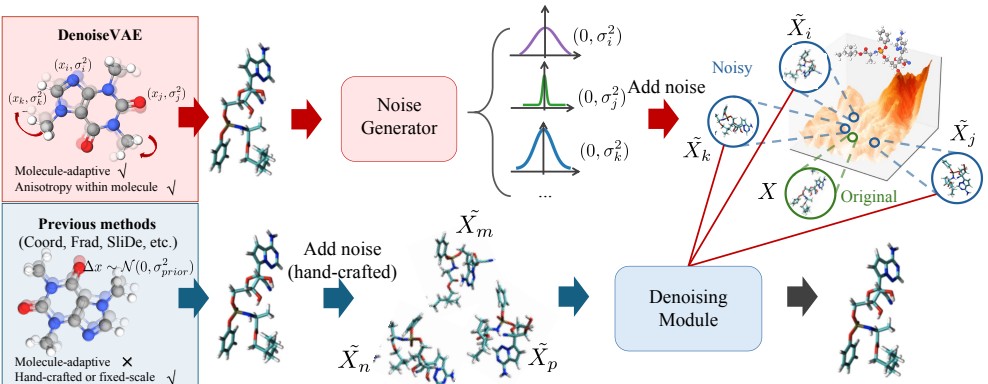

Figure 2: The overview of the comparison between our method and hand-crafted scale-invariant noise-based method (Zaidi et al., 2022; Feng et al., 2023; Ni et al., 2023). Different from denoising after adding hand-crafted noise, our method adopts a Noise Generator-Denoising Module paradigm, which generates atom-specific noise distributions through a trainable Noise Generator, and then samples the noises and reconstructs the original conformations. Our reconstruction task is equivalent to learning the force fields.

perturbing principle due to the difference in PES among different molecules. As shown in Fig. 2, in our method, we use a Noise Generator to generate an atom-specific noise distribution for each atom in different molecules. The noisy conformation is formed by a set of noisy atoms, each sampled from its corresponding distribution. This noisy conformation is then fed into the Denoising Module to produce an output. We use the reconstruction loss to guide the learning process of both the Denoising Module and the Noise Generator. In addition, we keep the similarity between the generated distribution and the pre-designed prior distribution, to avoid the Noise Generator from converging to a trivial solution. Compared with previous methods, we introduce a learnable module named Noise Generator, which is jointly trained with the Denoising Module. We provide the pseudocode in the Appendix A.6.

Formally, we define $\boldsymbol{X} = \{\boldsymbol{x}_1, \cdots, \boldsymbol{x}_N\}$, where $\boldsymbol{X} \in \mathbb{R}^{3 \times N}$ and $\boldsymbol{x}_i \in \mathbb{R}^3$ denote the original conformation and the position of the $i$-th atom in the molecule, respectively, and $N$ is the number of total atoms in the molecule. To obtain the corresponding noise for each atom, we model the Noise Generator $\mathcal{G}_{\boldsymbol{\varphi}}$, which takes the original molecular conformation $\boldsymbol{X}$ as input and generates a corresponding noise distribution for each atom within the molecule, as shown in Eq. 4. $\boldsymbol{\varphi}$ denotes the parameters of the Noise Generator.

$$\{\mathcal{N}(\boldsymbol{x}_1, \boldsymbol{\sigma}_{\boldsymbol{x}_1}^2), \cdots, \mathcal{N}(\boldsymbol{x}_N, \boldsymbol{\sigma}_{\boldsymbol{x}_N}^2)\} = \mathcal{G}_{\boldsymbol{\varphi}}(\boldsymbol{X}). \tag{4}$$

The noisy atom $\tilde{\boldsymbol{x}}_i$ should be randomly sampled from the corresponding generated distribution, where $\tilde{\boldsymbol{x}}_i \sim \mathcal{N}(\boldsymbol{x}_i, \boldsymbol{\sigma}_{\boldsymbol{x}_i}^2)$. Following Eq. 5, we achieve our sampling by applying the reparameterization, where $\boldsymbol{\epsilon}_{\boldsymbol{x}_i}$ represents random noise sampled from Standard Gaussian distribution.

$$\tilde{\boldsymbol{x}}_i = \boldsymbol{x}_i + \boldsymbol{\epsilon}_{\boldsymbol{x}_i} \boldsymbol{\sigma}_{\boldsymbol{x}_i}. \tag{5}$$

According to previous research (Vincent, 2011), the denoising task is equivalent to learning the molecular force fields. Therefore, we establish a Denoising Module $\mathcal{D}_{\boldsymbol{\theta}}$ to reconstruct the original molecular conformation, where $\boldsymbol{\theta}$ denotes the corresponding parameters. This module takes the perturbed molecular conformation $\tilde{\boldsymbol{X}}$, consisting of all noise-perturbed atoms $\{\tilde{\boldsymbol{x}}_1, \cdots, \tilde{\boldsymbol{x}}_N\}$, as input and predicts the corresponding noise for each atom. Through the Denoising Module, we acquire the output and denote it as $\mathcal{D}_{\boldsymbol{\theta}}(\tilde{\boldsymbol{X}}) = \{\hat{\boldsymbol{x}}_1, \cdots, \hat{\boldsymbol{x}}_N\}$. Following the recommendations from previous studies, we perform scaling operations using the atom-specific variance to obtain the loss for reconstructing the original molecular conformation, $\mathcal{L}_{\text{Denoise}}$, as follows:

$$\mathcal{L}_{\text{Denoise}} = \mathbb{E}_{p(\tilde{\boldsymbol{X}}, \boldsymbol{X})} \sum_{i=1}^{N} \boldsymbol{\sigma}_{\boldsymbol{x}_i}^2 \Big\| \hat{\boldsymbol{x}}_i - \frac{(\boldsymbol{x} - \tilde{\boldsymbol{x}})}{\boldsymbol{\sigma}_{\boldsymbol{x}_i}^2} \Big\|^2. \tag{6}$$

However, optimizing Eq.6 will lead the model to converge to a trivial solution, preventing the learning of meaningful noise. Therefore, we introduce a prior assumption, which is defined as follows:

$$\mathcal{L}_{\text{KL}} = \frac{1}{N} \sum_{i=1}^{N} D_{\text{KL}}\big(\mathcal{N}(0, \boldsymbol{\sigma}_{\boldsymbol{x}_i}^2) \| p_{\boldsymbol{x}_i}\big), \tag{7}$$

where $p_{\boldsymbol{x}_i}$ is the prior distribution for atom $\boldsymbol{x}_i$. Therefore, our final optimization objective for jointly training the Noise Generator and the Denoising Module is shown in Eq. 8,

$$\mathcal{L}_{\text{DenoiseVAE}} = \mathcal{L}_{\text{Denoise}} + \lambda \mathcal{L}_{\text{KL}}, \tag{8}$$

where $\lambda$ denotes the controlling hyper-parameter. In Eq. 8, the denoising objective constrains the scales of the generated molecule-adaptive noise distributions. If the applied noises exceed the physically permissible spheres of atomic activities under their structural contexts, the original conformation cannot be reconstructed. The KL divergence-based regularization term avoids generating extremely small noises for all atoms and increases the diversity of the sampled conformations.

## 4 THEORETICAL ANALYSIS UPON RATIONALE BEHIND DENOISEVAE

Our pretraining process constitutes a Variational Auto Encoder (VAE). Under our modeling approach, optimizing the pretraining objective is equivalent to improving the evidence lower bound (ELBO) of the log-likelihood. Below, we provide a detailed proof of this.

As shown in Fig. 3, we compare the energy changes in molecules before and after applying noise under three different noise addition modes. We randomly sample 10k molecules from the pretraining dataset and use sMAPE (Flores, 1986) for measurement. It can be observed that as the scale of the added noise decreases, the energy change of the molecule also significantly diminishes. Although our DenoiseVAE is constrained by a prior distribution with a larger variance than the Fixed scale-0.001, it can sample conformations with lower energy. Since the energy is closely related to molecular properties, our method successfully field conformations that most closely approximate the equilibrium molecular properties under different noise addition modes, thus reinforcing our original design intention. We formulate the phenomena in the following assumption.

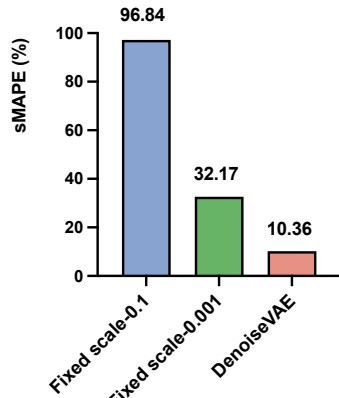

**Assumption 1** *Applying a limited noise perturbation $\delta$ to a molecule $X$ essentially leaves the molecular energy $E$ unchanged:*

$$E(X + \delta) \doteq E(X). \tag{9}$$

For DenoiseVAE, our goal is to better approximate $p(X)$, which represents the distribution of molecular conformations that essentially preserve the equilibrium state energy. Here, $\tilde{X}$ denotes the molecular conformation after noise perturbation, $\mathcal{M}$ denotes the pre-training dataset. Considering the ELBO of the log-likelihood $p(X)$:

Figure 3: Comparison of molecular potential energy changes before and after noise perturbation under different modes. Fixed scale-0.1 and Fixed scale-0.001 denote the noises are sampled from $\mathcal{N}(0, 0.1)$ and $\mathcal{N}(0, 0.001)$ for all atoms, respectively. For our DenoiseVAE, we set our prior distribution to $\mathcal{N}(0, 0.1)$.

$$\sum_{X_i \in \mathcal{M}} \log p(X_i) \geq \sum_{X_i \in \mathcal{M}} \Big( \mathbb{E}_{q_\varphi(\tilde{X}_i | X_i)} \Big[ \log p_\theta(X_i \mid \tilde{X}_i) \Big] - D_{\text{KL}}(q_\varphi(\tilde{X}_i \mid X_i) \| p_\theta(\tilde{X}_i)) \Big), \tag{10}$$

where $q_\varphi(\tilde{X}_i \mid X_i)$ represents the process of inputting the original molecular conformation into the Noise Generator to obtain the corresponding perturbed noisy molecule, $p_\theta(\tilde{X}_i)$ follows our predefined prior distribution, and $p_\theta(X_i \mid \tilde{X}_i)$ denotes the denoising process by which the Denoising Module removes noise from the perturbed molecule.

Our pretraining objective is to minimize the reconstruction loss $-\mathbb{E}_{q_\varphi(\tilde{X}_i|X_i)}\left[\log p_\theta(X_i \mid \tilde{X}_i)\right]$ with a KL divergence term as described in Eq. 7, which is equivalent to maximizing the ELBO of the log-likelihood. Moreover, under the assumption of the Boltzmann distribution (Boltzmann, 1868), our reconstruction process is equivalent to learning the molecular force field (Vincent, 2011), which in turn is equivalent to approximating the true distribution of molecular energy. Therefore, Eq. 10 can be rewritten as:

$$\sum_{X_i \in \mathcal{M}} \log p(X_i) \geq - \sum_{X_i \in \mathcal{M}} \mathcal{L}_{\text{DenoiseVAE}}. \tag{11}$$

When the pre-training process ends, we ultimately achieves an optimized posterior distribution, denoted as $\tilde{q}_\varphi(\tilde{X}_i \mid X_i)$. Due to better alignment with physical principles, our approach demonstrates a higher evidence lower bound (ELBO) of $p(X_i)$ compared to classical methods that directly sample noise from a Gaussian distribution. That is:

**Theorem 1** *Let $L(\tilde{q}_\varphi, p_\theta; X_i)$ denote the evidence lower bound (ELBO) of $p(X_i)$ obtained from our method, $L(p_\theta; X_i)$ denote the ELBO of $p(X_i)$ acquired from benchmark Gaussian-based methods. We derive:*

$$\mathbb{E}_{X_i \in \mathcal{M}}[L(\tilde{q}_\varphi, p_\theta; X_i)] \geq \mathbb{E}_{X_i \in \mathcal{M}}[L(p_\theta; X_i)]. \tag{12}$$

Theorem 1 indicates that our method provides a higher theoretical evidence lower bound guarantee for the real conformation distribution of isoenergetic molecules. Please refer to Appendix A.4 for the corresponding proofs.

## 5 EXPERIMENTS

### 5.1 SETTINGS

**Datasets** We leverage a large-scale molecular dataset **PCQM4Mv2** (Nakata & Shimazaki, 2017) as our pre-training dataset. It includes 3.4 million organic molecules, each represented by an equilibrium conformation and a label computed using density functional theory (DFT). We do not use this label in pre-training. For downstream tasks, we evaluate our method both on molecular and complex property prediction. For the former, we test on **QM9** (Ruddigkeit et al., 2012; Ramakrishnan et al., 2014), **MD17** (Chmiela et al., 2017) and **PCQM4Mv2** (Nakata & Shimazaki, 2017). For details, QM9 contains 12 chemical properties of small molecules with stable 3D structures. We follow previous work (Jiao et al., 2023) and split the dataset as the training set, validation set, and test set, which contains 100k, 18k, and 13k conformations, respectively. MD17 contains the simulated dynamical trajectories of 8 small organic molecules, with the recorded energy and force at each frame. We select 9,500 and 500 frames as the training and validation set respectively. PCQM4Mv2 (Nakata & Shimazaki, 2017) has a divided validation set and test set. We report the performance on the validation set according to formal standards, please refer to Appendix A.7 for details. For the latter, we adopt the widely recognized PDBBind dataset (v2019) for the ligand binding affinity (**LBA**) prediction, adhering to the 30% and 60% protein sequence identity splits and preprocessing methods outlined in Atom3D (Townshend et al., 2020).

**Experimental setup** We set the prior distribution $p_{\boldsymbol{x}_i}$ as a Gaussian distribution, where $\boldsymbol{x}_i$ is the mean and $\sigma$ is the standard deviation. If not specifically noted, we set $\sigma = 0.1$ for all experiments. After pre-training, we discard the Noise Generator and retain the Denoising Module for downstream tasks. It is worth mentioning that our DenoiseVAE is a lightweight network, and our entire pre-training process does not consume too much computing resources or time, which refers to the Appendix A.7. More details of our method are provided in the Appendix A.14.

Table 1: Performance (MAE ↓) on QM9 property prediction. The best results are in bold and the second best are underlined. For detailed standard deviation, please refer to Appendix A.8.

| Method | $\mu$ (D) | $\alpha$ ($a_0^3$) | $\epsilon_{HOMO}$ (meV) | $\epsilon_{LUMO}$ (meV) | $\Delta\epsilon$ (meV) | $<R^2>$ ($a_0^2$) | ZPVE (meV) | $U_0$ (meV) | $U$ (meV) | $H$ (meV) | $G$ (meV) | $C_v$ ($\frac{cal}{molK}$) |
|---|---|---|---|---|---|---|---|---|---|---|---|---|
| SchNet | 0.033 | 0.235 | 41.0 | 34.0 | 63.0 | 0.07 | 1.70 | 14.00 | 19.00 | 14.00 | 14.00 | 0.033 |
| E(n)-GNN | 0.029 | 0.071 | 29.0 | 25.0 | 48.0 | 0.11 | 1.55 | 11.00 | 12.00 | 12.00 | 12.00 | 0.031 |
| DimeNet++ | 0.030 | 0.044 | 24.6 | 19.5 | 32.6 | 0.33 | 1.21 | 6.32 | 6.28 | 6.53 | 7.56 | 0.023 |
| PaiNN | 0.012 | 0.045 | 27.6 | 20.4 | 45.7 | 0.07 | 1.28 | 5.85 | 5.83 | 5.98 | 7.35 | 0.024 |
| SphereNet | 0.025 | 0.045 | 22.8 | 18.9 | 31.1 | 0.27 | 1.120 | 6.26 | 6.36 | 6.33 | 7.78 | 0.022 |
| TorchMD-NET | 0.011 | 0.059 | 20.3 | 17.5 | 36.1 | **0.033** | 1.840 | 6.15 | 6.38 | 6.16 | 7.62 | 0.026 |
| Transformer-M | 0.037 | 0.041 | 17.5 | 16.2 | 27.4 | 0.075 | 1.18 | 9.37 | 9.41 | 9.39 | 9.63 | 0.022 |
| GeoSSL-DDM | 0.015 | 0.046 | 23.5 | 19.5 | 40.2 | 0.122 | 1.31 | 6.92 | 6.99 | 7.09 | 7.65 | 0.024 |
| 3D-EMGP | 0.020 | 0.057 | 21.3 | 18.2 | 37.1 | 0.092 | 1.38 | 8.60 | 8.60 | 8.70 | 9.30 | 0.026 |
| Coord | 0.012 | 0.0517 | 17.7 | 14.3 | 31.8 | 0.4496 | 1.71 | 6.57 | 6.11 | 6.45 | 6.91 | 0.020 |
| Frad | 0.010 | 0.0374 | 15.3 | 13.7 | 27.8 | 0.3419 | 1.418 | 5.33 | 5.62 | 5.55 | 6.19 | 0.020 |
| SliDe | 0.0087 | **0.0366** | **13.6** | 12.3 | 26.2 | 0.3405 | 1.521 | **4.28** | 4.29 | **4.26** | 5.37 | 0.019 |
| **DenoiseVAE** | **0.0079** | 0.0650 | 14.2 | **11.9** | **26.0** | 0.062 | **1.028** | 4.31 | **4.03** | **4.19** | 5.35 | **0.015** |

Table 2: Performance (MAE ↓) on MD17 force prediction. The best results are in bold and the second best are underlined. For detailed standard deviation, please refer to Appendix A.9.

| Method | Aspirin | Benzene | Ethanol | Malonaldehyde | Naphthalene | Salicylic Acid | Toluene | Uracil |
|---|---|---|---|---|---|---|---|---|
| TorchMD-NET | 0.1216 | 0.1479 | 0.0492 | 0.0695 | 0.0390 | 0.0655 | 0.0393 | 0.0484 |
| 3D-EMGP | 0.1560 | 0.1648 | 0.0389 | 0.0737 | 0.0829 | 0.1187 | 0.0619 | 0.0773 |
| 3D-EMGP (TorchMD-NET) | 0.1124 | 0.1417 | 0.0445 | 0.0618 | 0.0352 | 0.0586 | 0.0385 | 0.0477 |
| Coord | 0.0920 | 0.1397 | 0.0402 | 0.0661 | 0.0544 | 0.0790 | 0.0495 | 0.0507 |
| Frad | 0.0680 | 0.1606 | 0.0332 | **0.0427** | 0.0277 | **0.0410** | 0.0305 | **0.0323** |
| **DenoiseVAE** | **0.0567** | **0.1366** | **0.0303** | 0.1012 | **0.0219** | 0.1478 | **0.0301** | 0.0757 |

## 5.2 MAIN RESULTS

**QM9** We present the results on the QM9 dataset in Tab. 1. Our method achieves the best or the second best in 11 out of 12 tasks compared with other methods. For example, Frad uses hybrid noises to sample noisy conformations but still treats all molecules equally. Our method outperforms Frad in 11 out of 12 tasks, which indicates the importance of considering the PES differences among molecules and sampling the molecule-specific noises. Moreover, since we focus on sampling atom-specific and molecule-adaptive noise, the representations learned by our method exhibit a stronger ability to fit the dynamic properties of molecules. For example, we achieve a performance gain of **21.05%** in $C_v$ compared to SliDe, which represents the molecule's ability to absorb and store heat at different temperatures. This result exactly confirms that our method comprehensively learns the vibrational modes among and within molecules.

**MD17** The results are shown in Tab. 2. Our method achieves the best performance in 5 out of 8 tasks compared with other methods. For example, Coord samples noises from the same Gaussian distribution for all atoms in the molecules during the pre-training phase, and our method outperforms it on all the 8 tasks, which indicates the importance of taking the molecular anisotropy into account. It is worth mentioning that Frad focuses on modeling the anisotropy of the force fields by using hand-crafted hybrid noises but we still achieve **20.94%**, and **16.62%** performance gains on the Naphthalene, and Aspirin tasks compared with it. It proves that our atom-specific and molecule-adaptive noise sampling strategy is more reasonable, thus we can acquire a more generalizable representation for downstream tasks.

**LBA** We present the results in Tab. 3. Our method offers significant advantages over existing methods, especially at a sequence identity threshold of 30%, which consistently surpasses all baselines across various evaluation metrics, including RMSE, Pearson correlation, and Spearman correlation.

Table 3: Performance on LBA prediction. The best results are in bold and the second best are underlined. For detailed standard deviation, please refer to Appendix A.10.

| Method | Sequence Identity 30% | | | Sequence Identity 60% | | |
|---|---|---|---|---|---|---|
| | RMSE ↓ | Pearson ↑ | Spearman ↑ | RMSE ↓ | Pearson ↑ | Spearman ↑ |
| **Sequence Based** | | | | | | |
| DeepDTA | 1.866 | 0.472 | 0.471 | 1.762 | 0.666 | 0.663 |
| B&B | 1.985 | 0.165 | 0.152 | 1.891 | 0.249 | 0.275 |
| TAPE | 1.890 | 0.338 | 0.286 | 1.633 | 0.568 | 0.571 |
| ProtTrans | 1.544 | 0.438 | 0.434 | 1.641 | 0.595 | 0.588 |
| **Structure Based** | | | | | | |
| Holoprot | 1.464 | 0.509 | 0.500 | 1.365 | 0.749 | 0.742 |
| IEConv | 1.554 | 0.414 | 0.428 | 1.473 | 0.667 | 0.675 |
| MaSIF | 1.484 | 0.467 | 0.455 | 1.426 | 0.709 | 0.701 |
| ATOM3D-3DCNN | 1.416 | 0.550 | 0.553 | 1.621 | 0.608 | 0.615 |
| ATOM3D-ENN | 1.568 | 0.389 | 0.408 | 1.620 | 0.623 | 0.633 |
| ATOM3D-GNN | 1.601 | 0.545 | 0.533 | 1.408 | 0.743 | 0.743 |
| ProNet | 1.463 | 0.551 | 0.551 | **1.343** | 0.765 | **0.761** |
| **Pretraining Based** | | | | | | |
| GeoSSL | 1.451 | 0.577 | 0.572 | - | - | - |
| DeepAffinity | 1.893 | 0.415 | 0.426 | - | - | - |
| EGNN-PLM | 1.403 | 0.565 | 0.544 | 1.559 | 0.644 | 0.646 |
| Uni-Mol | 1.520 | 0.558 | 0.540 | 1.619 | 0.645 | 0.653 |
| **DenoiseVAE** | **1.401** | **0.588** | **0.574** | 1.497 | **0.769** | 0.746 |

At the 60% threshold, DenoiseVAE maintains its lead in the Pearson correlation and secures second place in the Spearman correlation. It is worth mentioning that since we pre-train on small molecules, our model is not supplemented by knowledge in the field of complex macromolecules. However, our method still performs better at the more stringent 30% threshold, which shows that DenoiseVAE has better robustness and generalization ability, especially when there is a significant difference between the training and test data distributions.

## 5.3 ABLATION STUDIES

**The influence of the hyper-parameter** $\sigma$    To prevent our Denoising Module from learning trivial solutions that result in meaningless noise, we introduce a prior distribution to constrain the model. As shown in Tab. 4, when the standard deviation of the prior distribution $\sigma$ is set to 0.1, our pre-trained model achieves optimal performance on both the QM9 and MD17 downstream tasks. This finding validates our initial intention behind the design of learnable noise: excessive or insufficient noise leads to suboptimal conformation sampling. If the noise is too large, molecular conformations deviate from the minima on the PES, resulting in the sampling of conformations with extremely low probabilities in quantum chemistry and molecular mechanics, such as saddle points or transition states. Conversely, if the noise is too small, the sampling search range of molecular conformations becomes too narrow, preventing adequate conformation sampling and potentially missing additional minima on the energy surface, i.e., stable conformations, leading to insufficient force field learning.

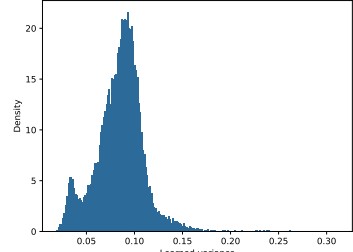

Figure 4: The distribution of the variances of our learned noise distributions.

As shown in Fig. 4, under the constraints of the prior distribution, our model adaptively samples from noise distributions with varying variances for atoms situated in different chemical environments.

**The influence of the hyper-parameter** $\lambda$    From the discussion in the previous section, we learn that the constraint on the prior part is crucial for our model. Therefore, we conduct an ablation study on the constraint strength $\lambda$ of this optimization objective. As shown in Tab. 5, when $\lambda$ is set to 1, our model performs satisfactorily on downstream tasks. This aligns with our Noise Generator-Denoising Module paradigm, where we encode the original molecular conformations into corresponding distributions and sample from them to reconstruct the original conformations.

Table 4: Performance (MAE ↓) with different $\sigma$.

| | QM9 | | MD17 | |
|---|---|---|---|---|
| | $\epsilon_{HOMO}$ | $\epsilon_{LUMO}$ | Aspirin | Benzene |
| $\sigma = 1$ | 23.6 | 16.9 | 0.2692 | 0.3883 |
| $\sigma = 0.5$ | 17.7 | 17.0 | 0.1423 | 0.1981 |
| $\sigma = 0.4$ | 15.8 | 12.1 | 0.0728 | 0.1725 |
| $\sigma = 0.1$ | **14.0** | **11.7** | **0.0559** | **0.1265** |
| $\sigma = 0.05$ | 26.5 | 22.4 | 0.1219 | 0.2927 |

Table 5: Performance (MAE ↓) with different $\lambda$.

| | QM9 | | MD17 | |
|---|---|---|---|---|
| | $\epsilon_{HOMO}$ | $\epsilon_{LUMO}$ | Aspirin | Benzene |
| $\lambda = 1$ | **14.0** | **11.7** | **0.0559** | **0.1265** |
| $\lambda = 0.5$ | 15.4 | 12.2 | 0.0722 | 0.1597 |
| $\lambda = 0.2$ | 18.7 | 20.6 | 0.2802 | 0.1773 |

## 5.4 CASE STUDIES

We select two typical examples for visual analysis. Our learnable noise strategy involves adaptive noise sampling for each atom within different molecules, as illustrated in Fig. 5. For both the two molecules, the noise intensity sampled by our model for each atom varies.

Specifically, the molecule in Fig. 5 (a) is an active chiral compound with polar functional groups, in which the amino group (-NH$_2$) can react with acids, the hydroxyl group (-OH) can form hydrogen bonds and react with acids or bases, and the cyano group (-CN) is polar and can participate in various organic reactions such as addition and reduction. For the molecule in Fig. 5 (b), its internal bicyclic structure endows it with high rigidity, making it less prone to structural changes. Therefore, compared to the molecule in Fig. 5 (a), the overall noise perturbation applied to it should be smaller.

Comparing the two molecules in Fig. 5, we can see that they contain some identical functional groups, such as hydroxyl groups. However, it is clear from Fig. 5 that the noise scale applied by our model to the oxygen atoms (O) in the hydroxyl groups of the two molecules differs. This further supports that our model

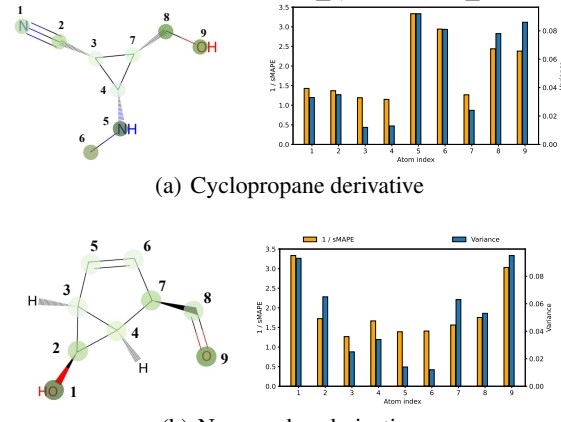

(a) Cyclopropane derivative

(b) Norcamphor derivative

Figure 5: Visualization of the intensity rationality of the learnable atom-level noise added to different molecules. A deeper green color indicates higher noise values. In the right tables, we follow the atom index order from the left molecular graphs and apply noises of the same scale to only one atom in the molecule at a time, calculating the overall energy change of the molecule before and after the perturbation.

adaptively performs meaningful sampling on the PESs of different molecules based on their dynamic characteristics. The two molecules presented in Fig. 5 exhibit significantly different properties. Through our design of learnable noise, we can adaptively sample conformations for different molecules, thereby learning better molecular representations and achieving superior performance in downstream tasks such as molecular property prediction.

## 6 CONCLUSION

In this paper, we explore the denoising method for 3D molecular pre-training. We propose an atom-specific and molecule-adaptive molecular pre-training method, named DenoiseVAE, which employs a Noise Generator to generate atom-specific noise distributions for different molecules and use a Denoising Module to reconstruct the original conformations. Since the sampling is more consistent with physical principles, our method can better capture the meaningful conformations of molecules, leading to more accurate force field learning which has a significant influence on downstream tasks. Furthermore, we provide a theoretical analysis of the rationale underlying DenoiseVAE, enhancing the interpretability of our method. The experimental results demonstrate the effectiveness of our method. It is worth noting that our method is based on classical force field assumptions. The potential integration with more accurate force fields represents a promising direction for future research.

ACKNOWLEDGEMENTS

This work is supported in part by the National Natural Science Foundation of China No. 62376277 and No. 61976206, Doubao large model fund and the Beijing Nova Program (No. 20230484278).

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

# A APPENDIX

## A.1 POTENTIAL ENERGY SURFACE

The Born-Oppenheimer approximation (BO approximation, also known as the adiabatic approximation) is a widely used method for solving the quantum mechanical equations of systems that include both electrons and atomic nuclei (Woolley & Sutcliffe, 1977). According to the Born-Oppenheimer approximation, the energy of the molecular ground electronic state can be regarded as a function of the nuclear coordinates. In molecular mechanics, all defined functions can be considered as functions of the nuclear coordinates, and the energy changes of the system can be viewed as movements on a multidimensional surface (Köuppel et al., 1984; Butler, 1998; Pachucki, 2010; Niklasson, 2008). The molecular mechanics force field function is shown in Eq. 13.

$$V(r^N) = \sum_{bond} \frac{k_i}{2}(l_i - l_{i,0})^2 + \sum_{angels} \frac{k_i}{2}(\theta_i - \theta_{i,0})^2 + \sum_{torsion} \frac{V_n}{2}(1 + \cos(n\omega - \gamma))$$
$$+ \sum_{i=1}^{N} \sum_{j=i+1}^{N} \left[ 4\varepsilon_{ij}\left(\left(\frac{\sigma_{ij}}{r_{ij}}\right)^{12} - \left(\frac{\sigma_{ij}}{r_{ij}}\right)^6\right) + \frac{q_i q_j}{4\pi\varepsilon_0 r_{ij}} \right]. \tag{13}$$

Therefore, the Potential Energy Surface (PES) of a molecule is a hypersurface formed by the potential energy with respect to all possible positions of the atoms. The positions of all atoms can be represented by a six-degree-of-freedom Cartesian coordinate system comprising translations and rotations.

On PES of the molecule, all minima are the primary focus of our work. At these points, moving in any direction on the PES, even with slight perturbations, will cause changes in the molecular structure, resulting in an increase in the molecular potential energy. The molecular PES may have multiple minima, each corresponding to an equilibrium structure of the system. For the same molecule, different minima on the PES correspond to different conformations or structural isomers. In a reaction system, these minima correspond to reactants, intermediates, products, and so on. Therefore, considering that the minima represent the true properties of the system, our work focuses on these points for study.

In the fields of quantum chemistry and molecular dynamics, to rigorously determine whether a given molecular conformation is sampled from a true minima on the PES, a frequency analysis is required, and the calculated frequencies should all be positive. If negative frequencies appear, it may be due to the constraints imposed by molecular symmetry. This occurs because if a molecule deviates from a minima position, it will experience a restoring force in the opposite direction, which allows the calculation of the molecule's vibrational frequencies. These vibrational frequencies correspond to the molecular spectrum.

For the stable conformations of molecules, we can employ several theoretical methods to obtain them. If the molecule is not very large, we can perform a conformation search by rotating all rotatable chemical bonds and then searching for the lowest energy points. Molecular dynamics simulations and other methods can also be used to search for conformations. However, if the molecule is very large, the aforementioned methods may not be able to find the true minimum; they can only find approximate local minima. Our work aims to provide solutions to the high computational cost of sampling equilibrium and stable conformations, exploring and attempting approaches that benefit practical drug discovery and new material design.

## A.2 MOLECULAR FORCE FIELD LEARNING

According to the prior knowledge in the field of statistical physics, the probability of the occurrence of 3D molecular conformation is described by Boltzmann distribution (Boltzmann, 1868).Therefore, we can obtain Eq. 14 through calculation.

$$\nabla_{\tilde{x}} \log p(\tilde{x}) = -\nabla_{\tilde{x}} E(\tilde{x}), \tag{14}$$

where $\tilde{x}$ denotes the perturbed conformation.

Learning the conformational distribution of molecules is equivalent to learning the molecular force field. Due to the lack of energy and force field labels, we generally learn approximate molecular

force fields by adding noise to the molecules and performing denoising tasks. Previous research has proven the equivalence of denoising tasks and learning molecular force fields (Vincent, 2011), as shown in Eq. 15. More details are shown in Sec. A.3 in the Appendix.

$$\nabla_{\tilde{x}} \log p(\tilde{x}|x_i) = -\frac{\tilde{x} - x_i}{\tau^2}, \tag{15}$$

where $\tau$ is a constant which can be absorbed into a graph neural network (GNN) (Feng et al., 2023). Previous work has proved this (Feng et al., 2023), as shown in Eq. 16.

$$
\begin{aligned}
&E_{p(\tilde{x})}||GNN_\theta(\tilde{x}) - (-\nabla_{\tilde{x}} E(\tilde{x}))||^2 \\
=&E_{p(\tilde{x})}||GNN_\theta(\tilde{x}) - \nabla_{\tilde{x}} \log p(\tilde{x})||^2 \\
=&E_{p(\tilde{x},x_i)}||GNN_\theta(\tilde{x}) - \nabla_{\tilde{x}} \log p(\tilde{x}|x_i)||^2 + T \\
=&E_{p(\tilde{x},x_i)}||GNN_\theta(\tilde{x}) - \frac{\tilde{x} - x_i}{\tau^2}||^2 + T.
\end{aligned}
\tag{16}
$$

## A.3 EQUIVALENCE OF DENOISING AND FORCE FIELD LEARNING

Since we have emphasized many times before that the denoising task is equivalent to learning the molecular force field, here we prove this hypothesis. The proof process refers to Frad (Feng et al., 2023).

*Proof* $J_1(\theta) = J_2(\theta)$.

$$J_1(\theta) = E_{p(\tilde{x})}||GNN_\theta(\tilde{x}) - \nabla_{\tilde{x}} \log p(\tilde{x})||^2, \tag{17}$$

$$J_2(\theta) = E_{p(\tilde{x}|x)p(x)}||GNN_\theta(\tilde{x}) - \nabla_{\tilde{x}} \log p(\tilde{x}|x)||^2. \tag{18}$$

$$J_1(\theta) = E_{p(\tilde{x})}[||GNN_\theta(\tilde{x})||^2] - 2E_{p(\tilde{x})}[< GNN_\theta(\tilde{x}), \nabla_{\tilde{x}} \log p(\tilde{x}) >] + T_3, \tag{19}$$

$$J_2(\theta) = E_{p(\tilde{x}|x)p(x)}[||GNN_\theta(\tilde{x})||^2] - 2E_{p(\tilde{x}|x)p(x)}[< GNN_\theta(\tilde{x}), \nabla_{\tilde{x}} \log p(\tilde{x}|x) >] + T_4, \tag{20}$$

$$
\begin{aligned}
&E_{p(\tilde{x})}[< GNN_\theta(\tilde{x}), \nabla_{\tilde{x}} \log p(\tilde{x}) >] \\
=&\int_{\tilde{x}} p(\tilde{x}) < GNN_\theta(\tilde{x},), \nabla_{\tilde{x}} \log p(\tilde{x}) > d\tilde{x} \\
=&\int_{\tilde{x}} p(\tilde{x}) \langle GNN_\theta(\tilde{x}), \frac{\nabla_{\tilde{x}} p(\tilde{x})}{p(\tilde{x})} \rangle d\tilde{x} \\
=&\int_{\tilde{x}} < GNN_\theta(\tilde{x}), \nabla_{\tilde{x}} p(\tilde{x}) > d\tilde{x} \\
=&\int_{\tilde{x}} \langle GNN_\theta(\tilde{x}), \nabla_{\tilde{x}} \big( \int_x p(\tilde{x}|x) p(x) dx \big) \rangle d\tilde{x} \\
=&\int_{\tilde{x}} \langle GNN_\theta(\tilde{x}), \int_x p(x) \nabla_{\tilde{x}} p(\tilde{x}|x) dx \rangle d\tilde{x} \\
=&\int_{\tilde{x}} \langle GNN_\theta(\tilde{x}), \int_x p(\tilde{x}|x) p(x) \nabla_{\tilde{x}} \log p(\tilde{x}|x) dx \rangle d\tilde{x} \\
=&\int_{\tilde{x}} \int_x p(\tilde{x}|x) p(x) < GNN_\theta(\tilde{x}), \nabla_{\tilde{x}} \log p(\tilde{x}|x) > dx d\tilde{x} \\
=&E_{p(\tilde{x},x)}[< GNN_\theta(\tilde{x}), \nabla_{\tilde{x}} \log p(\tilde{x}|x) >].
\end{aligned}
\tag{21}
$$

A.4  PROOF OF THE EVIDENCE LOWER BOUND OF DENOISEVAE

*Proof of Theorem 1*

$$logp(X_i) = \log p(X_i) \cdot \int q_\varphi(\tilde{X}_i|X_i) \, d\tilde{X}_i$$

$$= \int q_\varphi(\tilde{X}_i|X_i) \cdot \log p(X_i) \, d\tilde{X}_i \tag{22}$$

$$\log p(X_i) = \log \frac{p(X_i, \tilde{X}_i)}{p(\tilde{X}_i|X_i)}$$

$$= \log p(X_i, \tilde{X}_i) - \log p(\tilde{X}_i|X_i) \tag{23}$$

$$= \log \frac{p(X_i, \tilde{X}_i)}{q_\varphi(\tilde{X}_i|X_i)} - \log \frac{p(\tilde{X}_i|X_i)}{q_\varphi(\tilde{X}_i|X_i)}$$

Multiply both sides of the above formula by $q_\varphi(\tilde{X}_i|X_i)$ and take the integral:

$$\log p(X_i) = \int q_\varphi(\tilde{X}_i|X_i) \cdot \log \frac{p(X_i, \tilde{X}_i)}{q_\varphi(\tilde{X}_i|X_i)} \, d\tilde{X}_i - \int q_\varphi(\tilde{X}_i|X_i) \cdot \log \frac{p(\tilde{X}_i|X_i)}{q_\varphi(\tilde{X}_i|X_i)} \, d\tilde{X}_i$$

$$= D_{\text{KL}}(q_\varphi(\tilde{X}_i|X_i)\|p(\tilde{X}_i|X_i)) + \int q_\varphi(\tilde{X}_i|X_i) \cdot \log \frac{p_\theta(X_i|\tilde{X}_i)p_\theta(\tilde{X}_i)}{q_\varphi(\tilde{X}_i|X_i)} \, d\tilde{X}_i$$

$$= D_{\text{KL}}(q_\varphi(\tilde{X}_i|X_i)\|p(\tilde{X}_i|X_i)) + \int q_\varphi(\tilde{X}_i|X_i) \cdot \log p_\theta(X_i|\tilde{X}_i) \, d\tilde{X}_i \tag{24}$$

$$+ \int q_\varphi(\tilde{X}_i|X_i) \cdot \log \frac{p_\theta(\tilde{X}_i)}{q_\varphi(\tilde{X}_i|X_i)} \, d\tilde{X}_i$$

$$= D_{\text{KL}}(q_\varphi(\tilde{X}_i|X_i)\|p(\tilde{X}_i|X_i)) + \mathbb{E}_{\tilde{X}_i \sim q_\varphi(\tilde{X}_i|X_i)}[\log p_\theta(X_i|\tilde{X}_i)]$$

$$- D_{\text{KL}}(q_\varphi(\tilde{X}_i|X_i)\|p_\theta(\tilde{X}_i))$$

Since any KL divergence is non-negative, we can derive:

$$\log p(X_i) \geq \mathbb{E}_{\tilde{X}_i \sim q_\varphi(\tilde{X}_i|X_i)}[\log p_\theta(X_i|\tilde{X}_i)] - D_{\text{KL}}(q_\varphi(\tilde{X}_i|X_i)\|p_\theta(\tilde{X}_i)) \tag{25}$$

Let $L(\tilde{q}_\varphi, p_\theta; X_i)$ denote the evidence lower bound (ELBO) of $p(X_i)$ obtained from our method, $L(p_\theta; X_i)$ denote the ELBO of $p(X_i)$ acquired from benchmark classical methods which directly sample noise from a Gaussian distribution. That is:

$$L(\tilde{q}_\varphi, p_\theta; X_i) = \mathbb{E}_{\tilde{X}_i \sim q_\varphi(\tilde{X}_i|X_i)}[\log p_\theta(X_i|\tilde{X}_i)] - D_{\text{KL}}(q_\varphi(\tilde{X}_i|X_i)\|p_\theta(\tilde{X}_i)) \tag{26}$$

Our pretraining objective is to minimize the reconstruction loss $-\mathbb{E}_{q_\varphi(\tilde{X}_i|X_i)}\left[\log p_\theta(X_i \mid \tilde{X}_i)\right]$ with a KL divergence term as described in Eq. 7, which is equivalent to maximizing the ELBO of the log-likelihood. When the pre-training process ends, we ultimately achieves an optimized posterior distribution, denoted as $\tilde{q}_\varphi(\tilde{X}_i \mid X_i)$. Other benchmark Gaussian-based methods approximate the true noise distribution by directly using a Gaussian distribution. In contrast, our approach learns the noise distribution from the model based on a Gaussian prior. As a result, the noise we learn is more flexible and adaptive, aligning better with the physical principle of anisotropy in force fields. Consequently, the learned distribution $\tilde{q}_\varphi(\tilde{X}_i \mid X_i)$ is expected to be closer to the true noise distribution compared to other classical methods.

Let $\text{ELBO}_{noiseD}$ denote the ELBO of the true noise distribution $p_{noise}(X_i)$ obtained from our method, $\text{ELBO}_{noiseG}$ denote the ELBO of $p_{noise}(X_i)$ acquired from benchmark Gaussian-based methods. We have:

$$\text{ELBO}_{noiseD} \geq \text{ELBO}_{noiseG} \tag{27}$$

Therefore, the ELBO of $p(X_i)$ of our method is also higher than that of benchmark Gaussian-based methods. That is:

---

**Algorithm 1** Algorithm of our Denoise VAE

---

**Require:**
1: $\mathcal{D}_{\boldsymbol{\theta}}$: Denoising Module
2: $\mathcal{G}_{\boldsymbol{\varphi}}$: Noise Generator
3: $T$:Training steps
4: $S$: Training set
5: $p_{\boldsymbol{x}_i}$: Prior distribution
6: $\mathcal{N}$: Gaussian distribution
**Ensure:** $\mathcal{D}_{\theta}$.
7: **for** $i$ in $I$ **do**
8:     **for** $\boldsymbol{X}$ in $S$ **do**
9:         $[\boldsymbol{x}_1, \cdots, \boldsymbol{x}_N] = \boldsymbol{X}$
10:         $\{\mathcal{N}(\boldsymbol{x}_1, \boldsymbol{\sigma}_{\boldsymbol{x}_1}), \cdots, \mathcal{N}(\boldsymbol{x}_N, \boldsymbol{\sigma}_{\boldsymbol{x}_N})\} = \mathcal{G}_{\boldsymbol{\varphi}}(\boldsymbol{X})$
11:         **for** $\boldsymbol{x}$ in $\boldsymbol{X}$ **do**
12:             $\tilde{\boldsymbol{x}}_i = \boldsymbol{x}_i + \boldsymbol{\epsilon}_{\boldsymbol{x}_i} * \boldsymbol{\sigma}_{\boldsymbol{x}_i}$
13:         **end for**
14:         $\tilde{\boldsymbol{X}} = [\tilde{\boldsymbol{x}}_1, \cdots, \tilde{\boldsymbol{x}}_N]$
15:         $\{\hat{\boldsymbol{x}}_1, \cdots, \hat{\boldsymbol{x}}_N\} = \mathcal{D}_{\boldsymbol{\theta}}(\tilde{\boldsymbol{X}})$
16:         $\mathcal{L}_{\text{Denoise}} = E_{p(\tilde{\boldsymbol{X}}, \boldsymbol{X})} \sum_{i=1}^{N} \boldsymbol{\sigma}_{\boldsymbol{x}_i}^2 \left\| \hat{\boldsymbol{x}}_i - \frac{(\boldsymbol{x} - \tilde{\boldsymbol{x}})}{\boldsymbol{\sigma}_{\boldsymbol{x}_i}^2} \right\|^2$
17:         $\mathcal{L}_{\text{KL}} = \frac{1}{N} \sum_{i=1}^{N} D_{\text{KL}}\big(\mathcal{N}(\boldsymbol{x}_i | \boldsymbol{\mu}_{\boldsymbol{x}_i}, \boldsymbol{\sigma}_{\boldsymbol{x}_i}) || p_{\boldsymbol{x}_i}\big)$
18:         Optimize the object $\mathcal{L}_{\text{Denoise}} + \lambda \mathcal{L}_{\text{KL}}$
19:     **end for**
20: **end for**
21: **return** $\mathcal{D}_{\boldsymbol{\theta}}$

---

$$L(\tilde{q}_{\varphi}, p_{\theta}; X_i) \geq L(p_{\theta}; X_i) \tag{28}$$

Extending from one molecule to the entire dataset $\mathcal{M}$, we have

$$\mathbb{E}_{X_i \in \mathcal{M}}[L(\tilde{q}_{\varphi}, p_{\theta}; X_i)] \geq \mathbb{E}_{X_i \in \mathcal{M}}[L(p_{\theta}; X_i)]. \tag{29}$$

## A.5 THE PROBABILISTIC INVARIANCE OF DENOISEVAE

Our method satisfies O(3) probabilistic invariance, which also provides an explanation for the good performance of our method. We give the corresponding proof below:

Given $p(x'|x) = \mathcal{N}(x'|x, \sigma^2 I)$, we need to prove $p(Qx' + t|Qx + t) = p(x'|x)$, for $\forall Q \in O(3)$, $t \in \mathbb{R}^3$.

*Proof*

$$
\begin{aligned}
& p(Qx' + t|Qx + t) \\
=& \mathcal{N}(Qx' + t|Qx + t, \sigma^2 I) \\
=& \mathcal{N}(Qx'|Qx, \sigma^2 I) \\
=& \mathcal{N}(x'|x, \sigma^2 Q^T Q) \\
=& \mathcal{N}(x'|x, \sigma^2 I)
\end{aligned}
\tag{30}
$$

## A.6 PSEUDOCODE OF OUR METHOD

In order to show our training process more clearly, here we give the pseudocode of our entire method pipeline, as shown in Alg. 1.

## A.7 PERFORMANCE ON PCQM4Mv2

Due to the limitations in the length of the main paper, we present the performance of our method on the PCQM4Mv2 dataset here. As shown in Tab. 6, our method achieves optimal performance on the

validation set. Notably, despite having significantly fewer parameters compared to the other methods listed in the table, our model still demonstrates superior performance. This strongly validates the effectiveness of our approach.

In addition, we introduce the PCQM4Mv2 dataset in more details here. The PCQM4Mv2 dataset, part of the Open Graph Benchmark (OGB), is a large-scale quantum chemistry dataset designed for molecular property prediction and graph representation learning. It contains approximately 3.4 million organic molecules, each represented as molecular graphs with detailed atomic and bonding features. The primary task is to predict the HOMO-LUMO gap, a crucial property in quantum chemistry that reflects molecular stability and electronic behavior, with the target values derived from Density Functional Theory (DFT) calculations. The dataset is divided into training, validation, and test sets, supporting both supervised and pre-training tasks. PCQM4Mv2 is particularly valuable for training graph neural networks (GNNs) due to its scale and high-quality annotations. Its application spans self-supervised pretraining, molecular property prediction, and transfer learning. Despite its computationally demanding nature, it serves as a benchmark for models aiming to generalize across large and diverse molecular datasets.

Table 6: Performance on the PCQM4Mv2 dataset. The best results are in bold and the second best are underlined. We perform 3 replicates on the dataset to obtain the mean and standard deviation.

| Method | #param. | Valid MAE↓ |
|---|---|---|
| MLP-Fingerprint | 16.1M | 0.1735 |
| GCN | 2.0M | 0.1379 |
| GIN | 3.8M | 0.1195 |
| GINE$_{VN}$ | 13.2M | 0.1167 |
| GCN$_{VN}$ | 4.9M | 0.1153 |
| GIN$_{VN}$ | 6.7M | 0.1083 |
| DeeperGCN$_{VN}$ | 25.5M | 0.1021 |
| GraphGPS$_{SMALL}$ | 6.2M | 0.0938 |
| TokenGT | 48.5M | 0.0910 |
| GRPE$_{BASE}$ | 46.2M | 0.0890 |
| EGT | 89.3M | 0.0869 |
| GRPE$_{LARGE}$ | 46.2M | 0.0867 |
| Graphormer | 47.1M | 0.0864 |
| GraphGPS$_{BASE}$ | 19.4M | 0.0858 |
| GraphGPS$_{DEEP}$ | 13.8M | 0.0852 |
| GEM-2 | 32.1M | 0.0793 |
| GPS++ | 44.3M | 0.0778 |
| Transformer-M | 47.1M | 0.0787 |
| **DenoiseVAE** | **1.44M** | **0.0777 ± 0.0005** |

## A.8 PERFORMANCE ON QM9

For QM9, we compare our method with SchNet (Schütt et al., 2017), E(n)-GNN (Satorras et al., 2021), DimeNet++ (Gasteiger et al., 2020), PaiNN (Schütt et al., 2021), SpererNet (Liu et al., 2022b), TorchMD-NET (Thölke & De Fabritiis, 2022), Transformer-M (Luo et al., 2022), GeoSSL-DDM (Liu et al., 2022a), 3D-EMGP (Jiao et al., 2023), Coord (Zaidi et al., 2022), Frad (Feng et al., 2023), SliDE (Ni et al., 2023).

In order to verify the stability of our method, we randomly select 3 random seeds and run 3 times on the QM9 dataset to obtain the mean and standard deviation of our method, as shown in Tab. 7.

Moreover, we supply more information about the QM9 dataset here. The QM9 dataset is a widely used benchmark for quantum chemistry tasks, providing high-quality data for the prediction of molecular properties. It includes 134,000 small organic molecules, each containing up to 9 heavy atoms from elements H, C, N, O, and F. The dataset is derived from exhaustive quantum mechanical calculations using Density Functional Theory (DFT), ensuring accurate molecular descriptors. QM9 offers 12 molecular properties, including dipole moment, isotropic polarizability, HOMO/LUMO

Table 7: Performance (MAE ↓) on QM9 property prediction. The best results are in bold and the second best are underlined. Standard deviation is in parentheses.

| Method | $\mu$ (D) | $\alpha$ $(a_0^3)$ | $\epsilon_{HOMO}$ (meV) | $\epsilon_{LUMO}$ (meV) | $\Delta\epsilon$ (meV) | $<R^2>$ $(a_0^2)$ | ZPVE (meV) | $U_0$ (meV) | $U$ (meV) | $H$ (meV) | $G$ (meV) | $C_v$ $(\frac{cal}{molK})$ |
|---|---|---|---|---|---|---|---|---|---|---|---|---|
| SchNet | 0.033 | 0.235 | 41.0 | 34.0 | 63.0 | 0.07 | 1.70 | 14.00 | 19.00 | 14.00 | 14.00 | 0.033 |
| E(n)-GNN | 0.029 | 0.071 | 29.0 | 25.0 | 48.0 | 0.11 | 1.55 | 11.00 | 12.00 | 12.00 | 12.00 | 0.031 |
| DimeNet++ | 0.030 | 0.044 | 24.6 | 19.5 | 32.6 | 0.33 | 1.21 | 6.32 | 6.28 | 6.53 | 7.56 | 0.023 |
| PaiNN | 0.012 | 0.045 | 27.6 | 20.4 | 45.7 | 0.07 | 1.28 | 5.85 | 5.83 | 5.98 | 7.35 | 0.024 |
| SphereNet | 0.025 | 0.045 | 22.8 | 18.9 | 31.1 | 0.27 | 1.120 | 6.26 | 6.36 | 6.33 | 7.78 | 0.022 |
| TorchMD-NET | 0.011 | 0.059 | 20.3 | 17.5 | 36.1 | **0.033** | 1.840 | 6.15 | 6.38 | 6.16 | 7.62 | 0.026 |
| Transformer-M | 0.037 | 0.041 | 17.5 | 16.2 | 27.4 | 0.075 | 1.18 | 9.37 | 9.41 | 9.39 | 9.63 | 0.022 |
| GeoSSL-DDM | 0.015 | 0.046 | 23.5 | 19.5 | 40.2 | 0.122 | 1.31 | 6.92 | 6.99 | 7.09 | 7.65 | 0.024 |
| 3D-EMGP | 0.020 | 0.057 | 21.3 | 18.2 | 37.1 | 0.092 | 1.38 | 8.60 | 8.60 | 8.70 | 9.30 | 0.026 |
| Coord | 0.012 | 0.0517 | 17.7 | 14.3 | 31.8 | 0.4496 | 1.71 | 6.57 | 6.11 | 6.45 | 6.91 | 0.020 |
| Frad | 0.010 | 0.0374 | 15.3 | 13.7 | 27.8 | 0.3419 | 1.418 | 5.33 | 5.62 | 5.55 | 6.19 | 0.020 |
| SliDe | 0.0087 | **0.0366** | **13.6** | 12.3 | 26.2 | 0.3405 | 1.521 | **4.28** | 4.29 | 4.26 | 5.37 | 0.019 |
| **DenoiseVAE** | **0.0079** (0.00008) | 0.0650 (0.002) | 14.2 (0.12) | **11.9** (0.07) | **26.0** (0.24) | 0.062 (0.0007) | 1.028 (0.002) | 4.31 (0.009) | **4.03** (0.02) | **4.19** (0.03) | 5.35 (0.04) | **0.015** (0.0001) |

energy levels, and heat capacity, covering essential features for chemical and physical modeling. Each molecule is represented as a molecular graph, making it particularly suitable for developing and benchmarking graph neural networks (GNNs). With its standardized structure and focus on small molecules, QM9 is widely adopted for tasks such as molecular property prediction, representation learning, and generative modeling in computational chemistry and materials science.

## A.9 PERFORMANCE ON MD17

For MD17, we compare our method with TorchMD-NET (Thölke & De Fabritiis, 2022), 3D-EMGP (Jiao et al., 2023), 3D-EMGP(TorchMD-NET), Coord (Zaidi et al., 2022), and Frad (Feng et al., 2023).

In order to verify the stability of our method, we randomly select 3 random seeds and run 3 times on the MD17 dataset to obtain the mean and standard deviation of our method, as shown in Tab. 8.

Regarding the MD17 dataset, it is a benchmark for molecular dynamics and force field prediction tasks, providing time-series data of molecular geometries and corresponding energy and force labels. Derived from high-accuracy Density Functional Theory (DFT) simulations, MD17 includes 8 small organic molecules, such as benzene, toluene, and ethanol, spanning diverse chemical structures. Each molecule's dataset contains thousands of molecular conformations sampled along dynamic trajectories, with energy values and atomic forces calculated for each conformation. The primary tasks involve predicting the potential energy and forces ($kcal\,mol^{-1}\mathring{A}^{-1}$) to model molecular interactions, enabling applications in force field learning and molecular dynamics simulations. Unlike static datasets like QM9, MD17 captures temporal dependencies, making it ideal for developing machine learning models that understand molecular behavior over time. The dataset challenges models to generalize across unseen conformations and accurately predict properties, providing a testbed for equivariant graph neural networks and other advanced architectures.

## A.10 PERFORMANCE ON LBA

For LBA, we compare our method with DeepDTA (Öztürk et al., 2018), B&B (Bepler & Berger, 2019), TAPE (Rao et al., 2019), ProtTrans (Elnaggar et al., 2021), Holoprot (Somnath et al.), IEConv (Hermosilla et al., 2020), MaSIF (Gainza et al., 2020), ATOM3D-3DCNN, ATOM3D-ENN, ATOM3D-GNN (Townshend et al., 2020), ProNet (Wang et al., 2022), GeoSSL (Liu et al., 2022a), EGNN-PLM (Wu et al., 2022), DeepAffinity (Karimi et al., 2018) and Uni-Mol (Zhou et al., 2023).

We run three times under the two splittings to obtain the corresponding mean and standard deviation. The experimental results are shown in Tab. 9 and Tab. 10, which further proves the stability of our method.

Table 8: Performance (MAE ↓) on MD17 force prediction. The best results are in bold and the second best are underlined. Standard deviation is in parentheses.

| Method | Aspirin | Benzene | Ethanol | Malonaldehyde | Naphthalene | Salicylic Acid | Toluene | Uracil |
|---|---|---|---|---|---|---|---|---|
| TorchMD-NET | 0.1216 | 0.1479 | 0.0492 | 0.0695 | 0.0390 | 0.0655 | 0.0393 | 0.0484 |
| 3D-EMGP | 0.1560 | 0.1648 | 0.0389 | 0.0737 | 0.0829 | 0.1187 | 0.0619 | 0.0773 |
| 3D-EMGP (TorchMD-NET) | 0.1124 | 0.1417 | 0.0445 | 0.0618 | 0.0352 | 0.0586 | 0.0385 | 0.0477 |
| Coord | 0.0920 | 0.1397 | 0.0402 | 0.0661 | 0.0544 | 0.0790 | 0.0495 | 0.0507 |
| Frad | 0.0680 | 0.1606 | 0.0332 | **0.0427** | 0.0277 | **0.0410** | 0.0305 | **0.0323** |
| **DenoiseVAE** | **0.0567** (0.006) | **0.1366** (0.011) | **0.0303** (0.008) | 0.1012 (0.015) | **0.0219** (0.001) | 0.1478 (0.018) | **0.0301** (0.007) | 0.0757 (0.006) |

Table 9: Performance of sequence identity 30% on LBA prediction with standard deviation.

| Method | Sequence Identity 30% | | |
|---|---|---|---|
| | RMSE ↓ | Pearson ↑ | Spearman ↑ |
| **Sequence Based** | | | |
| DeepDTA | 1.866±0.08 | 0.472±0.02 | 0.471±0.02 |
| B&B | 1.985±0.01 | 0.165±0.01 | 0.152±0.02 |
| TAPE | 1.890±0.04 | 0.338±0.04 | 0.286±0.12 |
| ProtTrans | 1.544±0.02 | 0.438±0.05 | 0.434±0.06 |
| **Structure Based** | | | |
| Holoprot | 1.464±0.01 | 0.509±0.00 | 0.500±0.01 |
| IEConv | 1.554±0.02 | 0.414±0.05 | 0.428±0.03 |
| MaSIF | 1.484±0.02 | 0.467±0.02 | 0.455±0.01 |
| ATOM3D-3DCNN | 1.416±0.02 | 0.550±0.02 | 0.553±0.01 |
| ATOM3D-ENN | 1.568±0.01 | 0.389±0.02 | 0.408±0.02 |
| ATOM3D-GNN | 1.601±0.05 | 0.545±0.03 | 0.533±0.03 |
| ProNet | 1.463±0.00 | 0.551±0.01 | 0.551±0.01 |
| **Pretraining Based** | | | |
| GeoSSL | 1.451±0.03 | 0.577±0.02 | 0.572±0.01 |
| DeepAffinity | 1.893±0.65 | 0.415 | 0.426 |
| EGNN-PLM | 1.403±0.01 | 0.565±0.02 | 0.544±0.01 |
| Uni-Mol | 1.520±0.03 | 0.558±0.00 | 0.540±0.00 |
| **DenoiseVAE** | **1.401±0.01** | **0.588±0.03** | **0.574±0.01** |

In addition, the LBA (Ligand Binding Affinity) dataset is a benchmark for predicting the binding affinity between protein-ligand complexes, a critical task in drug discovery and computational biology. It contains 4,463 protein-ligand complex structures, each annotated with experimental binding affinity values in units of -log(Kd), derived from the PDBBind database. Each complex includes detailed 3D spatial information of the protein and ligand, capturing key structural and chemical interactions. The primary goal is to predict binding affinity, helping assess molecular docking accuracy and protein-ligand interaction modeling. LBA's diverse dataset spans a range of protein sizes, ligand structures, and binding strengths, making it suitable for evaluating models' generalization across chemical and structural variations. It is widely used for training and benchmarking graph neural networks (GNNs) and deep learning models incorporating 3D spatial features. LBA is instrumental for advancing computational approaches in precision medicine and protein-ligand interaction analysis.

## A.11 VISUALIZATION OF ENERGY LANDSCAPE

To investigate the learned representation space of our pre-trained model, we visualize the local energy landscape around a given molecular conformation sampled from MD17 dataset. Specifically, following 3D-EMGP (Jiao et al., 2023), we select a random conformation $X$ from the dataset and utilize the pre-trained Noise Generator to obtain the atom-specific noise distributions for $X$. According to the noise distributions, we randomly generate two directions $G_1, G_2 \in \mathbb{R}^{3 \times N}$. Then we

Table 10: Performance of sequence identity 60% on LBA prediction with standard deviation.

| Method | Sequence Identity 60% | | |
|---|---|---|---|
| | RMSE ↓ | Pearson ↑ | Spearman ↑ |
| **Sequence Based** | | | |
| DeepDTA | 1.762±0.26 | 0.666±0.01 | 0.663±0.02 |
| B&B | 1.891±0.00 | 0.249±0.01 | 0.275±0.01 |
| TAPE | 1.633±0.02 | 0.568±0.03 | 0.571±0.02 |
| ProtTrans | 1.641±0.02 | 0.595±0.01 | 0.588±0.01 |
| **Structure Based** | | | |
| Holoprot | 1.365±0.04 | 0.749±0.01 | 0.742±0.01 |
| IEConv | 1.473±0.02 | 0.667±0.01 | 0.675±0.02 |
| MaSIF | 1.426±0.02 | 0.709±0.01 | 0.701±0.00 |
| ATOM3D-3DCNN | 1.621±0.03 | 0.608±0.02 | 0.615±0.03 |
| ATOM3D-ENN | 1.620±0.05 | 0.623±0.02 | 0.633±0.02 |
| ATOM3D-GNN | 1.408±0.07 | 0.743±0.02 | 0.743±0.03 |
| ProNet | **1.343±0.03** | 0.765±0.01 | **0.761±0.00** |
| **Pretraining Based** | | | |
| EGNN-PLM | 1.559±0.02 | 0.644±0.02 | 0.646±0.02 |
| Uni-Mol | 1.619±0.04 | 0.645±0.02 | 0.653±0.02 |
| **DenoiseVAE** | 1.497±0.02 | **0.769±0.01** | 0.746±0.01 |

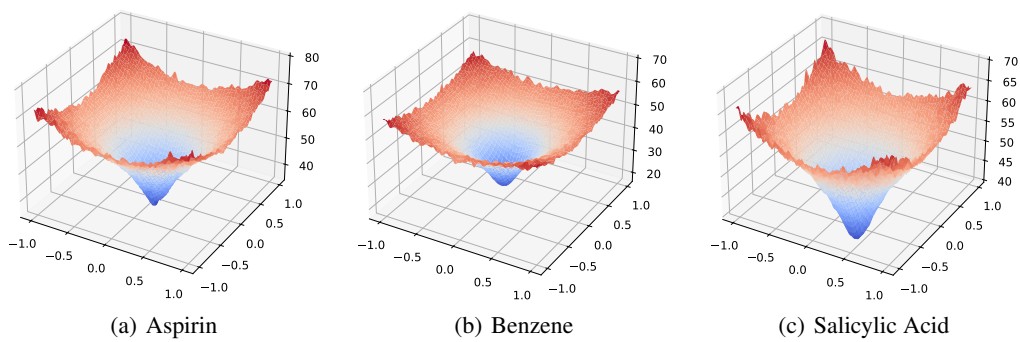

(a) Aspirin        (b) Benzene        (c) Salicylic Acid

Figure 6: Visualization of energy landscape on MD17 dataset.

construct a 2D conformation plane as $\{\tilde{\boldsymbol{X}}(i,j)|\tilde{\boldsymbol{X}}(i,j) = \boldsymbol{X} + i\boldsymbol{G_1} + j\boldsymbol{G_2}\}$. For each point in the plane, by varying $i$ and $j$, we calculate the corresponding energy as $E_{i,j} = E(\tilde{\boldsymbol{X}}(i,j))$, where E denotes the energy calculation function in RDKit library (Landrum, 2006). As shown in Fig. 6, we select Aspirin, Benzene, and Salicylic Acid to plot the energy landscape $(i,j,E_{i,j})$. It is evident that the molecular force field learned by our model can identify the equilibrium conformation as a local energy minimum point on the energy landscape. Moreover, the energy surfaces of different molecules vary greatly. Among them, the energy surface of the Benzene converges to its steady-state conformation at a significantly slower rate than the Salicylic Acid, which is consistent with real chemical constraints and also verifies our motivation.

## A.12  VISUALIZATION OF THE STABILITY OF OUR METHOD

Our method demonstrates strong training stability during both the pretraining and finetuning stages, achieving consistent convergence under various initialization and optimization settings. Since model optimization is primarily influenced by data initialization, learning rate, and the number of iterations—and the learning rates used by existing methods in the field are largely consistent (Zaidi et al., 2022; Feng et al., 2023; Ni et al., 2023), we validate the stability of our method by setting different random seeds to vary data initialization and visualizing the loss at different iteration steps. As shown in Fig. 7, whether during pre-training on the PCQM4Mv2 dataset or supervised fine-tuning

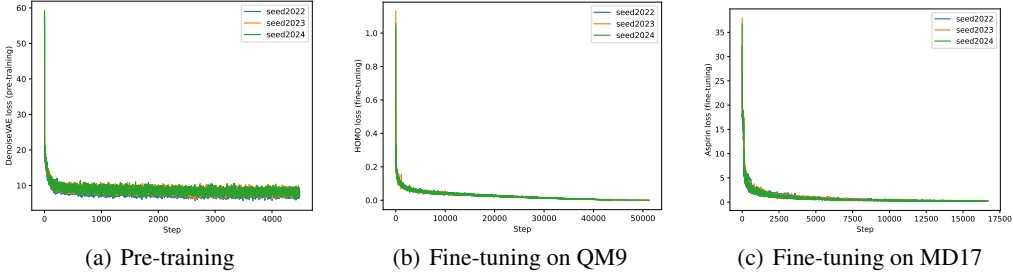

Figure 7: Loss curves of our method.

on downstream datasets such as QM9 and MD17, the loss of our method converges quickly to a relatively low value. This provides strong evidence of the stability of our method.

### A.13 MORE ABLATION STUDIES

As shown in Tab. 11, we perform ablation studies on the number of Equivariant Graph Neural Network (EGNN) layers used by the Noise Generator in our DenoiseVAE. We find that when the number of layers is 4, our model performs better on downstream tasks. This shows that appropriately increasing the depth of the model can learn better molecular representations, but at the same time, it will bring about an increase in computing resources.

Moreover, we conduct an ablation study on the noise sampling strategies. Our method is designed to focus solely on learning the diagonal elements of the covariance matrix and perform independent noise sampling for each atom. Here, we extend the model to account for atomic relationships by learning the full covariance matrix, thereby capturing the dependencies between different atoms within the molecule when sampling the noises. Formally, we denote $\boldsymbol{X} \in \mathbb{R}^{3 \times N}$, where $N$ is the number of atoms in the molecule $\boldsymbol{X}$. We use the Noise Generator $\mathcal{G}_\phi$ to model the noise distribution for each atom in a given molecule. That is:

$$\mathcal{N}(\boldsymbol{X}, \boldsymbol{\Sigma}) = \mathcal{G}_{\boldsymbol{\varphi}}(\boldsymbol{X}), \tag{31}$$

where $\boldsymbol{\Sigma} \in \mathbb{R}^{N \times N}$. The noisy molecular conformation $\tilde{\boldsymbol{X}}_i$ should be sampled from the corresponding generated distribution. Similarly, we achieve our sampling by applying reparameterization as follows, where $\epsilon$ represents random noise sampled from the Standard Gaussian distribution.

$$\tilde{\boldsymbol{X}} = \boldsymbol{X} + \boldsymbol{\epsilon}\boldsymbol{\Sigma}^{\frac{1}{2}}. \tag{32}$$

As shown in Eq. 33 and Eq. 34, we compare the three noise sampling strategies in different methods. $\boldsymbol{\Sigma}_{i,i}$ and $\boldsymbol{\Sigma}_{i,j}$ denote the noise variance of the $i$-th atom and covariance between the $i$-th and the $j$-th atoms. 'Traditional method' refers to adding pre-defined and constant noise to all atoms for each molecule. 'Ours (independent)' represents our DenoiseVAE method, in which our model learns the noise distribution for each atom individually and performs independent noise sampling during the sampling phase. 'Ours (non-independent)' highlights an enhancement of our DenoiseVAE, where inter-atomic relationships are incorporated by considering the off-diagonal elements of the covariance matrix, $\boldsymbol{\Sigma}_{i,j}$, rather than setting them directly to zero. In fact, our method is a subcase of this design.

$$\boldsymbol{\Sigma}_{i,i} = \begin{cases} \sigma^2 & \text{, Traditional method} \\ \sigma_i^2 & \text{, Ours (independent)} \\ \sigma_{i,i}^2 & \text{, Ours (non-independent)} \end{cases} \tag{33}$$

$$\boldsymbol{\Sigma}_{i,j} = \begin{cases} 0 & \text{, Traditional method} \\ 0 & \text{, Ours (independent)} \\ \sigma_{i,j}^2 & \text{, Ours (non-independent)} \end{cases} \tag{34}$$

Through the Denoising Module, we acquire the output and denote it as $\hat{\boldsymbol{X}} = \mathcal{D}_{\boldsymbol{\theta}}(\tilde{\boldsymbol{X}})$. The final denoising loss is shown as follows, where $\boldsymbol{X}_i$ and $\hat{\boldsymbol{X}}_i$ denote the $i$-th atom in the molecule $X$ and

output $\hat{\boldsymbol{X}}$, respectively. $\boldsymbol{\Sigma}_{i,i}$ represents the corresponding noise variance of $i$-th atom.

$$\mathcal{L}_{\text{Denoise}} = \mathbb{E}_{p(\tilde{\boldsymbol{X}}, \boldsymbol{X})} \sum_{i=1}^{N} \boldsymbol{\Sigma}_{i,i} \big\| \hat{\boldsymbol{X}}_i - \frac{(\boldsymbol{X}_i - \tilde{\boldsymbol{X}}_i)}{\boldsymbol{\Sigma}_{i,i}} \big\|^2. \tag{35}$$

Since directly optimizing the denoising loss will lead the model to a trivial solution, we introduce a prior distribution to binding the learned distribution as follows:

$$\mathcal{L}_{\text{KL}} = D_{\text{KL}}\big(\mathcal{N}(0, \boldsymbol{\Sigma}) \| p_{\boldsymbol{X}}\big), \tag{36}$$

where $p_{\boldsymbol{X}}$ is the pre-defined distribution. Then the final optimization target is:

$$\mathcal{L}_{\text{DenoiseVAE}} = \mathcal{L}_{\text{Denoise}} + \lambda \mathcal{L}_{\text{KL}}. \tag{37}$$

The experimental results are shown in Tab. 12. We observe that the experimental results under these two noise sampling settings are comparable. This may be attributed to the fact that our Noise Generator is an equivariant graph neural network, which takes into account and integrates information from neighboring atoms around each atom during the pre-training process. Although independent perturbations are applied to each atom during the final noise sampling stage, the scale of these perturbations is, in fact, highly correlated with the surrounding atoms.

Table 11: Ablation studies on the number of EGNN layers of the Noise Generator.

|  | QM9 | | MD17 | |
| --- | --- | --- | --- | --- |
|  | $\epsilon_{HOMO}$ | $\epsilon_{LUMO}$ | Aspirin | Benzene |
| layer=4 | **14.0** | **11.7** | **0.0559** | **0.1265** |
| layer=2 | 15.6 | 12.0 | 0.0761 | 0.1589 |

Table 12: Ablation studies on the noise sampling strategy (atom-independent/non-independent).

|  | QM9 | | | | MD17 | | | |
| --- | --- | --- | --- | --- | --- | --- | --- | --- |
|  | $\mu$ | $\alpha$ | $\epsilon_{HOMO}$ | $\epsilon_{LUMO}$ | Aspirin | Benzene | Ethanol | Malonaldehyde |
| **Ours** (non-independent) | 0.0082 | **0.0639** | 14.5 | 12.2 | 0.0590 | 0.1389 | **0.0302** | **0.0989** |
| **Ours** (independent) | **0.0079** | 0.0650 | **14.2** | **11.9** | **0.0567** | **0.1366** | 0.0303 | 0.1012 |

A.14 DETAILS ABOUT EXPERIMENTAL SETTINGS

We present the details about the hyper-parameters of our experiments in Tab. 13.

For training resources, all experiments are conducted on Intel(R) Xeon(R) Gold 5318Y CPU @ 2.10GHz with a single RTX A3090 GPU. Normally, 6 GPUs with a total of 144GB of memory is sufficient for the reproduction.

Table 13: Hyper-parameters for Pre-training dataset.

| Dataset | PCQM4Mv2 |
| --- | --- |
| Batch size | 128 |
| Optimizer | AdamW |
| Max learning rate | 0.0005 |
| Learning rate decay policy | Cosine |
| Network archecture | Equivariant Graph Neural (EGNN) |
| Noise Generator layers | 4 |
| Denoising Module layers | 7 |

