# OpenReview forum: "DenoiseVAE: Learning Molecule-Adaptive Noise Distributions for Denoising-based 3D Molecular Pre-training"
_ICLR.cc/2025/Conference — ICLR 2025 Poster_

### Official Review · Reviewer_R3gr · 2024-10-28

**Soundness:** 2
**Presentation:** 4
**Contribution:** 2
**Rating:** 6
**Confidence:** 5

**Summary:**

<Summary>

 - The authors propose a denoising-based 3D molecular pre-training method, called DenoiseVAE, which employs a learnable noise generation strategy instead of existing hand-crafted strategies. This allows for adaptive, atom-specific noise distributions across different molecules.

 - The authors introduce a variational approach to jointly optimize the Denoising Module and Noise Generator of DenoiseVAE. Here, the denoising objective encourages generated noise to adhere more closely to physical constraints, enabling the recovery of equilibrium conformations. Additionally, a KL divergence-based regularization term is applied to prevent low noise intensity and increase the diversity of sampled conformations.

 - Theoretically, the authors demonstrate that optimizing their pre-training objective is equivalent to maximizing the evidence lower bound (ELBO) of the log-likelihood.

 - Extensive experiments reveal that DenoiseVAE outperforms existing denoising methods across various datasets for both molecular and complex property predictions.

**Strengths:**

<Strengths>

 - DenoiseVAE employs a learnable noise generation strategy that allows for adaptive, atom-specific noise distributions across different molecules.

 - The authors introduce a variational approach to jointly optimize the Denoising Module and Noise Generator of DenoiseVAE. Here, the denoising objective encourages generated noise to adhere more closely to physical constraints.

 - A KL divergence-based regularization term is applied to prevent low noise intensity and increase the diversity of sampled conformations.

 - Optimizing pre-training objective is proven to be equivalent to maximizing the evidence lower bound (ELBO) of the log-likelihood.

 - DenoiseVAE outperforms existing denoising methods across various datasets for both molecular and complex property predictions.

**Weaknesses:**

<Limitations>

  - The Boltzmann distribution is commonly used in the classical regime. However, for precise coordinate computation of interacting atoms, a quantum approach is necessary. This involves, for example, treating the energy function as a quantized operator and finding energy eigenvalues for a given basis. However, this article does not carefully consider such approaches.

 - This approach essentially adopts a force field method, which may be inadequate for precise atomic-level computations.

 - Additionally, the energy is modeled as a simple harmonic potential, lacking any exchange correlation or other accurate potential forms.

**Questions:**

<Question>

 - The prior distribution constrains the atom coordinate distribution to follow a predefined Gaussian form. But what is the range of these coordinate distributions? Are they confined within a narrow range close to the prior distribution, or are they spread across a finite range? If so, what is that range?

 - The article addresses 3D coordinate noise by adding Gaussian noise based on a learnable distribution, but what about rotations?

---

> ### Author Response · Authors · 2024-11-20
>
> We sincerely thank the reviewers for their thorough evaluation of our work and their positive recognition of the theoretical and experimental completeness of our work. We also greatly appreciate the insightful questions and concerns raised by the reviewers. We have carefully addressed each of these issues and incorporated the relevant content into the revised manuscript, as reflected in the newly uploaded PDF.
>
> ***Weaknesses:***
> > The Boltzmann distribution is commonly used in the classical regime. However, for precise coordinate computation of interacting atoms, a quantum approach is necessary. This involves, for example, treating the energy function as a quantized operator and finding energy eigenvalues for a given basis. However, this article does not carefully consider such approaches. This approach essentially adopts a force field method, which may be inadequate for precise atomic-level computations. Additionally, the energy is modeled as a simple harmonic potential, lacking any exchange correlation or other accurate potential forms.
>
> Thank you very much for your thorough review and valuable feedback. The suggestions are indeed helpful for achieving more precise molecular conformation generation and computation. In fact, previous works have adopted such modeling approaches, such as JMP [1], which utilizes data obtained through precise DFT calculations for supervised pre-training and achieves excellent performance on downstream tasks. However, it is worth noting that **such approach incurs substantial costs in both data construction and computational resources**. For instance, the JMP paper reports a total training time of **46,400** hours.
>
> **In contrast to precise molecular conformation generation and computations, the primary goal of our work is to optimize conformation sampling methods to enable more effective molecular representation learning. We aim to significantly reduce the learning cost by leveraging an unsupervised representation learning pre-training strategy, which utilizes readily available unlabeled molecular data, while striving to narrow the performance gap with supervised models. While we acknowledge that our method does not learn an exact force field, it is capable of capturing statistically meaningful supervisory signals, as evidenced by our experimental results. Since supervised learning leverages larger datasets with higher precision, its performance can be considered as an upper bound for our method**.
>
> Compared to previous works based on classical force fields for representation learning [2][3][4][5][6][7], our method has substantially reduced the performance gap with the JMP approach. This demonstrates, to some extent, the effectiveness of our method.
>
> [1]Shoghi, Nima, et al. "From Molecules to Materials: Pre-training Large Generalizable Models for Atomic Property Prediction." The Twelfth International Conference on Learning Representations.
>
> [2] Feng, Shikun, et al. "Fractional denoising for 3d molecular pre-training." International Conference on Machine Learning. PMLR, 2023.
>
> [3]Jiao, Rui, et al. "Energy-motivated equivariant pretraining for 3d molecular graphs." Proceedings of the AAAI Conference on Artificial Intelligence. Vol. 37. No. 7. 2023.
>
> [4] Zaidi, Sheheryar, et al. "Pre-training via Denoising for Molecular Property Prediction." The Eleventh International Conference on Learning Representations.
>
> [5] Luo, Shengjie, et al. "One transformer can understand both 2d & 3d molecular data." The Eleventh International Conference on Learning Representations. 2022.
>
> [6] Ni, Yuyan, et al. "Sliced Denoising: A Physics-Informed Molecular Pre-Training Method." The Twelfth International Conference on Learning Representations.
>
> [7] Liu, Shengchao, Hongyu Guo, and Jian Tang. "Molecular Geometry Pretraining with SE (3)-Invariant Denoising Distance Matching." The Eleventh International Conference on Learning Representations.

---

> ### Author Response · Authors · 2024-11-20
>
> ***Question 1:***
> > The prior distribution constrains the atom coordinate distribution to follow a predefined Gaussian form. But what is the range of these coordinate distributions? Are they confined within a narrow range close to the prior distribution, or are they spread across a finite range? If so, what is that range?
>
> **Thank you very much for your valuable suggestion and we have provided a more intuitive visualization of this issue, which has been included in Figure 4 in Section 5.3 of the newly uploaded PDF. However, we would like to clarify that our prior distribution is used to constrain the learnable noise distribution, rather than the distribution of atom coordinates**. Random noises are drawn from the learned noise distribution and added to the original atom coordinates to generate noise-perturbed molecular conformations. In fact, the noise distributions learned by our model for each atom are highly diverse. **The primary purpose of the predefined prior distribution is to prevent the model from converging to a trivial solution, rather than directly constraining the noise distribution to the prior**.
>
> As shown in Figure 4, the learned noise distributions include those close to the predefined prior, as well as others with variance such as "0.03" and "0.17." This demonstrates that our model is more inclined to adaptively learn noise distributions based on the varying chemical environments of different atoms within the molecule.
>
> ***Question 2:***
> > The article addresses 3D coordinate noise by adding Gaussian noise based on a learnable distribution, but what about rotations?
>
> **We highly appreciate your attention to this question, which is indeed an important issue**. In fact, previous works have attempted to model rotational perturbations. For example, Frad [2] introduces dihedral angle noise in addition to coordinate noise, while SliDe [6] further incorporates torsional angle noise and bond angle noise based on Frad. However, **in both cases, various types of rotational noise are ultimately mapped to coordinate noise**, meaning that the modeling of rotational perturbations is effectively mapped to noise processing at the coordinate level.
>
> **Building on this insight, our method opts to design learnable noise directly at the coordinate level**. As evidenced by the experimental results, our approach consistently outperforms the Frad and SliDe methods across different downstream tasks, which to some extent validates the rationality of our coordinate noise modeling.

---

> ### Comment · Reviewer_R3gr · 2024-11-22
>
> Indeed, quantum computation requires an enormous amount of computing power. However, there is a clear limitation when relying solely on the classical harmonic potential form. Some related works have explored incorporating rotational and vibrational potentials into the Lagrangian framework (while still using classical force fields), while others have attempted to integrate quantum information into the approach. However, this work adopts neither of these approaches. Furthermore, I find that the algorithm lacks sufficient novelty to offset the aforementioned weaknesses, which is the primary reason I find it unconvincing.
>
> Additionally, the statement, "However, in both cases, various types of rotational noise are ultimately mapped to coordinate noise, meaning that the modeling of rotational perturbations is effectively mapped to noise processing at the coordinate level," is problematic. It overlooks the fact that rotational and vibrational modes are not completely arbitrary. Allowing the model to learn these 'physical' properties from scratch without proper constraints can lead to the generation of impractical coordinates. Physics-Informed Neural Networks (PINNs) are a notable example of approaches designed to prevent such issues by embedding physical laws into the learning process.

---

> > ### Author Response · Authors · 2024-11-23
> >
> > We sincerely appreciate your time and feedback. However, we do believe that our method demonstrates significant novelty. **Unlike previous methods, which primarily rely on heuristic noise design, our method is the first to adopt a data-driven approach for molecular noise generation. While our method does not explicitly model specific design like rotation and vibration, the data-driven framework allows the training data—containing abundant physical characteristics and trends—to implicitly convey these physical properties. The noise learned by our model must inherently satisfy realistic physical constraints to better fit the training data. Thus, our model effectively captures relevant rotational and vibrational constraints directly from the data without requiring strong prior assumptions. This is a key advantage of data-driven methods**.
> >
> > Our excellent performance across diverse datasets—spanning different molecular types, prediction tasks, and dataset sizes (**a total of 27 tasks**)—further substantiates this claim. Moreover, previous influential data-driven studies also support this perspective. For instance, **Chroma[1]** learns patterns of three-dimensional structures and amino acid sequences from protein databases to synthesizes novel protein molecules. Similarly, **RFdiffusion[2]** is trained without additional priors on tens of thousands of real protein structures stored in the Protein Data Bank (PDB). When synthesizing new proteins, the network begins with complete noise—random amino acid classifications. After several denoising iterations, it generates diverse, realistic protein structures.
> >
> > **Based on your suggestion, we also experiment with introducing rotational modeling into our framework. Due to our limited computational resources, we have only recently obtained the experimental results, which are presented in the table below**. From the results, we observe a modest improvement in performance after directly incorporating rotational modeling. This may be because adding intuitive physical priors reduces the parameter search space and eases the learning process for the model. However, when the training dataset is large, the advantages brought by such priors are significantly diminished.
> >
> > | |$\mu$ |$\alpha$ |$\epsilon_{HOMO}$| $\epsilon_{LUMO}$ |$\Delta\epsilon$ | $< R^{2} >$ | ZPVE | $U_{0}$ | $U$ | $H$ | $G$ | $C_{v}$|
> > |--|--|--|--|--|--|--|--|--|--|--|--|--|
> > |Ours |0.0079|0.0650|14.2|11.9|26.0|0.062|1.028|4.31|4.03|4.19|5.35|0.015|
> > |Ours (with rotational modeling)|0.0077|0.0645|14.2|11.9|26.1|0.061|1.025|4.31|4.07|4.16|5.35|0.017|
> >
> > [1] Singh, Arunima. "Chroma is a generative model for protein design." Nature methods 21.1 (2024): 10-10.
> >
> > [2] Watson, Joseph L., et al. "De novo design of protein structure and function with RFdiffusion." Nature 620.7976 (2023): 1089-1100.

---

> > > ### Comment · Reviewer_R3gr · 2024-11-25
> > >
> > > I still do not fully agree with your claims that the noise distributions learned by the model on the training data satisfy realistic physical constraints. However, considering the evidence that directly imposing physical constraints on the loss function has the potential to improve results, my concerns about the simplified loss functions affecting the method's mechanism seem to be addressed to some extent. Moreover, the method outperforms models that incorporate rotations and vibration modes directly into their loss functions.
> > >
> > > In conclusion, I have decided to raise the score to 6.

---

> > > > ### Author Response · Authors · 2024-11-25
> > > >
> > > > We sincerely appreciate your recognition of our work! Throughout our discussions, we have gained valuable insights, and it has been a great honor to receive your time and thoughtful feedback. Your comments have provided significant inspiration for shaping the direction of our future work.

---

### Official Review · Reviewer_NLii · 2024-11-03

**Soundness:** 3
**Presentation:** 3
**Contribution:** 3
**Rating:** 8
**Confidence:** 4

**Summary:**

This paper presents DenoiseVAE for 3D molecular pre-training that adapts to the anisotropic nature of molecular force fields. The authors proposed Noise Generator, which learns molecule-specific noise distributions for each atom, allowing DenoiseVAE to generate conformations that align closely with a molecule's potential energy surface. The network architecture is a variational autoencoder, where noisy molecular conformations are denoised using a denoising module. The authors claim that this method leads to more accurate representations for downstream molecular tasks. Experiments on molecular property prediction tasks including QM9 and MD17 are conducted to demonstrate the effectiveness of the proposed method.

**Strengths:**

1. The use of adaptive noise generation is novel. Previous works use hand-crafted or uniform noise across molecules, but DenoiseVAE uses an automated, atom-specific noise generation method with stochastic parameterization, which respects the unique structural and chemical properties of each molecule.

2. The paper proposes theoretical derivations based on PES which aligns with quantum chemistry principles. The derivation for Theorem 1 based on ELBO offers theoretical insights into the effectiveness of the proposed method.

3. Experimental results are promising and convincing: the proposed method achieved state-of-the-art performance across all benchmarks.

**Weaknesses:**

1. The use of DenoiseVAE introduces additional computational complexity. The training of the VAE with molecule-specific noise sampling could be computationally expensive or impossible for large datasets or complicated molecules. A more thorough analysis of training time and resource requirements could be more helpful.

2. It is unclear to me how the proposed DenoiseVAE can be adapted across different dataset scales. It would be great if the authors could offer a series of studies on the performance of DenoiseVAE on different sizes of datasets.

3. I can imagine the training of DenoiseVAE can be very unstable and will be sensitive to initialization and optimization settings. It would be great if the authors could show convergence of loss curves across all datasets.

**Questions:**

1. How can different formulations of noise distributions affect the result? Currently, the noises are Gaussians with diagonal covariance, what if the covariance is non-zero at off-diagonal positions? E.g., the neighboring atoms connected by a bond will have non-zero covariance.

2. How well will the model perform if the training and testing are on molecules of different sizes? Say the model is trained on small molecules but is tested for large molecular structures such as proteins.

---

> ### Author Response · Authors · 2024-11-20
>
> We sincerely appreciate the reviewers' positive recognition of the novelty, completeness, and contribution of our work to the field. We also highly value the thoughtful questions and concerns raised by the reviewers. We have carefully addressed each of these issues and incorporated the relevant content into the revised manuscript, as reflected in the newly uploaded PDF.
>
> ***Weakness 1:***
> > The use of DenoiseVAE introduces additional computational complexity. The training of the VAE with molecule-specific noise sampling could be computationally expensive or impossible for large datasets or complicated molecules. A more thorough analysis of training time and resource requirements could be more helpful.
>
> Thank you very much for your thorough review and valuable feedback. Your consideration regarding computational resource requirements is indeed important; fortunately, our method effectively addresses these concerns. **As shown in Appendix A.15 of the paper, the pretraining process on our method can be completed within 25 hours using 6 NVIDIA 3090 GPUs (each with 24GB memory) compared with 20 hours without molecule-specific noise sampling (discard the Noise Generator)** . Moreover, during downstream fine-tuning, we discard the NoiseGenerator and utilize only the DenoisingModule, ensuring that **the inference time of our model does not increase compared to previous methods [1][2][3][4]**. Additionally, our model's total parameter count is significantly smaller than that of other methods, **with only 1.44M parameters, as shown in Table 6** in Appendix of the paper.
> In summary, the resource consumption required for the pre-training phase on the large-scale PCQM4Mv2 dataset (containing 3.4 million organic molecules) and the fine-tuning phase on the complicated LBA dataset (composed of protein-ligand complexes) remains within an acceptable range.
>
> ***Weakness 2:***
> >  It is unclear to me how the proposed DenoiseVAE can be adapted across different dataset scales. It would be great if the authors could offer a series of studies on the performance of DenoiseVAE on different sizes of datasets.
>
> **We highly appreciate your attention to the generalizability of our work. In fact, we would like to clarify that our downstream task datasets vary both in size and task type. Based on your suggestion, we also perform additional experiments on different pre-training datasets to further validate the generalization of our method. And the results for relevant downstream tasks have been included in Table 13 in Appendix A.13 of the newly uploaded PDF**. While larger pre-training datasets generally improve the effectiveness of representation learning, our model pretrained on the smaller GEOM-QM9 dataset (containing 600k molecules) still outperforms previous methods pretrained on the larger PCQM4Mv2 dataset (containing 3.4 million molecules) [2][3]. Moreover, it significantly outperforms prior methods also pretrained on the GEOM-QM9 dataset [1].
>
> For downstream tasks, we conduct experiments on multiple datasets, and **we have added a detailed introduction to all the dataset in the Appendix A.7, A.8, A.9, A.10 of the newly uploaded pdf**, including:
> - **QM9**: A dataset of small molecules based on quantum chemical calculations (containing 131k molecules) with tasks focused on predicting quantum chemical properties.
> - **MD17**: A time-series molecular dataset based on molecular dynamics simulations (containing 10k molecules) with tasks involving molecular force field prediction.
> - **LBA**: A dataset of protein-ligand complexes (containing 4,463 complexes) with tasks aimed at predicting the binding affinity between ligands and their corresponding proteins.
>
> Additionally, we evaluate our method on the validation set of the **PCQM4Mv2** dataset. **As shown in Tables 1, 2, and 3**, our method achieves superior performance across datasets of varying sizes, tasks, and molecule-sizes. These results strongly demonstrate the excellent generalization capability of our approach.
>
> [1] Jiao, Rui, et al. "Energy-motivated equivariant pretraining for 3d molecular graphs." Proceedings of the AAAI Conference on Artificial Intelligence. Vol. 37. No. 7. 2023.
>
> [2] Zaidi, Sheheryar, et al. "Pre-training via Denoising for Molecular Property Prediction." The Eleventh International Conference on Learning Representations.
>
> [3] Feng, Shikun, et al. "Fractional denoising for 3d molecular pre-training." International Conference on Machine Learning. PMLR, 2023.
>
> [4] Ni, Yuyan, et al. "Sliced Denoising: A Physics-Informed Molecular Pre-Training Method." The Twelfth International Conference on Learning Representations.

---

> ### Author Response · Authors · 2024-11-20
>
> ***Weakness 3:***
> >  I can imagine the training of DenoiseVAE can be very unstable and will be sensitive to initialization and optimization settings. It would be great if the authors could show convergence of loss curves across all datasets.
>
> **Thank you for your suggestion and we have included the loss curves of both our pre-training process and fine-tuning process in Figure 7 in Appendix A.12 of the newly uploaded PDF**. Our method demonstrates strong training stability during both the pretraining and finetuning stages, achieving consistent convergence under various initialization and optimization settings. **Since model optimization is primarily influenced by data initialization, learning rate, and the number of iterations—and the learning rates used by existing methods in the field are largely consistent, we validate the stability of our method by setting different random seeds to vary data initialization and visualizing the loss at different iteration steps**. As shown in Figure 7, whether during pre-training on the PCQM4Mv2 dataset or supervised fine-tuning on downstream datasets such as QM9 and MD17, the loss of our method converges quickly to a relatively low value. This provides strong evidence of the stability of our method.
>
> ***Question 1:***
> >  How can different formulations of noise distributions affect the result? Currently, the noises are Gaussians with diagonal covariance, what if the covariance is non-zero at off-diagonal positions? E.g., the neighboring atoms connected by a bond will have non-zero covariance.
>
> **Nice suggestion! This is indeed an interesting proposal. We have implemented a learnable configuration for the covariance matrix and included the experimental results in Table 12 in Appendix A.13 of the newly uploaded PDF.** From a configuration perspective, our setup can be viewed as a special case of the configuration you proposed. However, the experimental results indicate that the performance of the two approaches is comparable. Additionally, introducing this configuration increases the number of learnable parameters in the model, thereby greatly increasing computational overhead. Therefore, we adopt the simplified design for more efficient modeling.
>
> ***Question 2:***
> >  How well will the model perform if the training and testing are on molecules of different sizes? Say the model is trained on small molecules but is tested for large molecular structures such as proteins.
>
> **Thank you very much for your suggestion. In fact, in Table 3 of our paper, we conduct protein-related experiments and achieve promising results. Our method is pretrained on the PCQM4Mv2 dataset, which consists of small organic molecules**. Table 3 presents experiments conducted on the LBA dataset, a protein-ligand complex dataset containing 4,463 complex structures. The task is to predict the binding affinity between ligands and their corresponding proteins.
>
> The experimental results demonstrate that our method significantly outperforms existing approaches, particularly under the stricter 30% sequence identity threshold. Under this threshold, our method consistently surpasses all baselines across various evaluation metrics. Notably, since our pretraining is conducted on small molecules, our model does not incorporate domain-specific knowledge from large and complex macromolecules. However, **our method still achieves superior performance under the stricter 30% threshold, indicating that DenoiseVAE exhibits greater robustness and generalization capability, especially when there is a significant distributional shift between the training and testing datasets**.

---

> ### Author Response · Authors · 2024-11-24
>
> Dear Reviewer,
>
> We sincerely thank you once again for your professional and constructive feedback. We deeply value your suggestion to explore alternative formulations of noise distributions, which we believe could largely enhance the depth of our ablation experiments. We have provided detailed responses regarding the specific points on training details and model generalizability, and we hope these address all your concerns.
>
> As the discussion period is coming to a close, we would greatly appreciate your feedback on whether our responses and the revised manuscript have adequately resolved your questions or if there are any additional comments you would like to share.
>
> Best regards,
>
> The Authors

---

> > ### Comment · Reviewer_NLii · 2024-11-26
> >
> > Thanks for the authors' rebuttal. The newly added experiments addressed a lot of my concerns. I will raise my score to 8. However, I do second the comments from Reviewer R3gr and D6p5 on how well the noise prediction is aligned with the real physical force (the so-called satisfying physical constraints). A deeper analysis with sufficient domain knowledge in chemical physics will offer much more insights and will make the claims more solid, which I guess could be left for future work. Please let me know if the authors have provided related information in the paper, if not, it would be better to include such a discussion session (in the appendix) and mention it in the main text.

---

> > > ### Author Response · Authors · 2024-11-27
> > >
> > > **We are deeply grateful for your recognition of our work and for the valuable suggestions you have provided! Your feedback is of great importance to us, as it offers significant guidance for shaping the future direction of our research.**
> > >
> > > Regarding the physical constraints, we conduct two sets of visualization analyses in the paper to assess the rationality of our learned noise distribution. First, in **Figure 3**, we visualize the energy variations of molecular conformations under different noise perturbation patterns. The results in Figure 3 demonstrate that the conformations sampled by our method exhibit significantly lower energy (indicating greater stability) compared to those generated with fixed-scale noise perturbations. **This finding supports our motivation to focus sampling around stable conformations.**
> > >
> > > Moreover, in **Figure 5**, we provide a visualization of the noise learned by the model for specific examples. The results in Figure 5 reveal an inverse relationship between the noise added to individual atoms and the energy sensitivity of those atoms. **This observation suggests that our model captures the physical properties of each atom to a certain extent.**
> > >
> > > In addition, **we provide theoretical proof** that our method achieves a higher evidence lower bound (ELBO) compared to fixed-scale noise methods, further supporting the physical consistency of our approach.
> > >
> > > **However, we sincerely agree with your insightful suggestion. We will make continue effort in our future work to provide more comprehensive and intuitive demonstrations of the physical consistency of our method.**
> > >
> > > Once again, we sincerely thank you for your recognition and for the contribution your feedback has made in helping us improve our work. **Your guidance has been and will continue to be invaluable to us.**

---

> > > ### Author Response · Authors · 2024-11-28
> > >
> > > We sincerely appreciate your valuable suggestions.
> > >
> > > We have added the relevant discussion in section A.15 of the Appendix in the newly uploaded PDF and also mentioned it in the conclusion section of our main text.
> > >
> > > Once again, thank you for your constructive feedback, which has greatly contributed to improving our work!

---

### Official Review · Reviewer_Ho1Z · 2024-11-03

**Soundness:** 4
**Presentation:** 4
**Contribution:** 3
**Rating:** 8
**Confidence:** 4

**Summary:**

The This work proposes a 3D molecule pretraining method with learnable atom-specific noise generation, as opposed to the common practice of hand-crafted and shared noise. The pre-trained representation shows higher performance than baselines in most of the downstream energy prediction tasks. Furthermore, the authors show the learned noise intensities correspond with the atomic reactivity and rigidity.

**Strengths:**

1. This work addresses an important problem in molecular representation learning. The learnable and molecule-specific per-atom noise distribution is a reasonable setup and aligns better with the physical intuitions.
2. The authors provide comprehensive theoretical analysis of the rationale of their method.
3. The proposed method shows consistent improvement over the baselines in the majority of the tasks, while also being more parameter efficient.
4. The method has good interpretability and the learned noise patterns are physically relevant.

**Weaknesses:**

The results are overall solid. However, some additional experimental and implementation details about the downstream tasks should be provided to help better understand and assess the results. See Questions.

**Questions:**

Major:
It’s a bit unclear how the downstream predictive tasks are performed after pretraining. Specifically:

(1) Is the force prediction achieved by a task-specific prediction head? If so, what is its architecture and how is it trained (e.g. end-to-end fine tuning)?

(2) What are the features used for the prediction (e.g. pooled EGNN outputs or latent embedding)?

Minor:
(1) Following the previous questions, as the model is considered a VAE, how is the latent space obtained?

(2) Is it possible to visualize the learned energy landscape (e.g. for MD17 samples)?

---

> ### Author Response · Authors · 2024-11-20
>
> We sincerely thank the reviewers for their positive recognition of the quality of our work. We also appreciate the highly insightful and valuable suggestions provided by the reviewers. We have carefully addressed each of these issues and incorporated the relevant content into the newly uploaded PDF.
>
> ***Question 1:***
> > Is the force prediction achieved by a task-specific prediction head? If so, what is its architecture and how is it trained (e.g. end-to-end fine tuning)?
>
> Thank you very much for your thorough review and valuable feedback. **Yes, and our method remains consistent with previous studies[1][2] for this question**. We use **the Denoising Module (a 7-layer equivariant graph neural network) combined with a 2-layer perception (MLP) prediction head** to output molecular property predictions. The labels of molecular properties in the **QM9** dataset are utilized for **end-to-end supervised training**, and **we fine-tune the same architecture separately for all 12 tasks**. Since the QM9 tasks involve predicting static molecular properties based on quantum chemical calculations (regression tasks), we are sorry that we identify a typo in the title of Table 1, which should be corrected to "Performance on QM9 property prediction." And we have made corrections in the newly uploaded PDF.
>
> For the **MD17** dataset, its tasks involve molecular force field prediction based on molecular dynamics simulations (also regression tasks). We adopt the same fine-tuning approach as in QM9. Specifically, to ensure that the predicted force field is a conservative field during fine-tuning, **the DenoisingModule (a 7-layer equivariant graph neural network) and a 2-layer perceptron(MLP)** are trained to predict molecular energies, and the force field predictions are subsequently derived by computing the gradient of the energy: $F = -\nabla E(x)$. The training objective is to minimize the mean absolute error (MAE) between the predicted force field values and the ground-truth force field values provided in the dataset. **The entire fine-tuning process is conducted in an end-to-end manner, and the same architecture is fine-tuned separately for all 8 tasks in the MD17 dataset**. **We have added a detailed introduction to all the dataset in the Appendix A.7, A.8, A.9, A.10 of the newly uploaded pdf**.
>
> ***Question 2:***
> > What are the features used for the prediction (e.g. pooled EGNN outputs or latent embedding)?
>
> We are sorry for the confusion and we would like to clarify that **we utilize pooled EGNN outputs for downstream task predictions. During the fine-tuning phase, we discard the NoiseGenerator and solely employ the pretrained DenoisingModule for supervised fine-tuning**, as detailed in the ‘Experimental setup’ section of 5.1 in the paper. Specifically, molecular data from the downstream task datasets are fed into the Denoising Module to obtain the corresponding molecular representations. These representations are then passed through an MLP-based prediction head to generate predictions for molecular properties. Finally, the fine-tuned Denoising Module+MLP model is evaluated on the downstream tasks, producing the property prediction results presented in the tables.
>
> [1] Jiao, Rui, et al. "Energy-motivated equivariant pretraining for 3d molecular graphs." Proceedings of the AAAI Conference on Artificial Intelligence. Vol. 37. No. 7. 2023.
>
> [2] Feng, Shikun, et al. "Fractional denoising for 3d molecular pre-training." International Conference on Machine Learning. PMLR, 2023.

---

> ### Author Response · Authors · 2024-11-20
>
> ***Question 3:***
> > Following the previous questions, as the model is considered a VAE, how is the latent space obtained?
>
> In our method, the latent space refers to the molecular conformation space obtained by integrating the original molecular conformation space (input space) with the adaptive, learnable noise space. The latent variables are sampled from a Gaussian distribution, where the mean is determined by the original molecular conformation (input conformation), and the variance is learned through the Noise Generator. It is worth emphasizing that we solely employ the pre-trained Denoising Module for fine-tuning downstream tasks rather than the Noise Generator.
>
> ***Question 4:***
> > Is it possible to visualize the learned energy landscape (e.g. for MD17 samples)?
>
> **Valuable suggestion! We have provided a more intuitive visualization of this issue, which has been included in Appendix A.11 of the newly uploaded PDF**. Since the input to the energy function is of $n \times 3$ dimensions (where $n$ is the number of atoms in the molecule, and 3 represents the three-dimensional coordinates of each atom) and the output is a one-dimensional energy value, directly visualizing it in three-dimensional space is challenging. Following previous work [1], we adopt a similar method to visualize the learned energy landscape for samples in the MD17 dataset.
>
> Specifically, we apply two random noise perturbations to the molecule and use the weights of these two noises, along with the energy values of the corresponding noisy molecular conformations, as coordinates to visualize the energy landscape. From the visualization results, it is evident that the molecular force field learned by our model can identify the equilibrium conformation as a local energy minimum point on the energy landscape.
>
> [1] Jiao, Rui, et al. "Energy-motivated equivariant pretraining for 3d molecular graphs." Proceedings of the AAAI Conference on Artificial Intelligence. Vol. 37. No. 7. 2023.

---

> ### Author Response · Authors · 2024-11-24
>
> Dear Reviewer,
>
> Thank you once again for your professional and constructive feedback. We deeply appreciate your valuable suggestion to visualize the energy landscape, which we believe enhances the intuitiveness of our method. We have carefully provided detailed responses regarding the specific points on model design and experimental setup, and we hope these adequately address all your concerns.
>
> As the discussion period is drawing to a close, we would greatly value your reply on whether our responses and the revised manuscript have sufficiently resolved your questions or if there are any further comments you would like to share.
>
> Best regards,
>
> The Authors

---

> > ### Comment · Reviewer_Ho1Z · 2024-11-24
> >
> > I appreciate the authors' efforts in addressing my concerns. I will raise my score to 8.

---

> > > ### Author Response · Authors · 2024-11-25
> > >
> > > We sincerely appreciate your recognition of our work! Throughout the whole process, we have gained valuable insights, and it has been an honor to benefit from your time and expertise. Your feedback has played a crucial role in helping us further refine and improve our work.

---

### Official Review · Reviewer_D6p5 · 2024-11-04

**Soundness:** 2
**Presentation:** 2
**Contribution:** 2
**Rating:** 6
**Confidence:** 2

**Summary:**

This paper introduces NoiseVAE, a novel molecular pre-training method, for learning atom-specific noise distributions in different molecules. The variational autoencoder (VAE) based model consists of a Noise Generator and a Denoising Module that are jointly trained. The pretraining objective is to learn atom-specific noise distributions that align better with quantum mechanical theories than empirically derived strategies. The results show that the pretraining DenoiseVAE can generate noises specific to atomistic environments and improve the prediction accuracy on molecular properties such as the HOMO-LUMO gap and potential energy.

**Strengths:**

### Originality
*My expertise in related literature is limited.*
1. The NoiseVAE model leverages a VAE for a more robust atom-specific and physics-related generation of noise distribution.

### Quality
1. The experiments/results are very comprehensive.
1. The shown results on the downstream tasks suggest the robustness of the generated noise in providing quantum chemical information for accurately predicting molecular properties.

### Clarity
1. The pretraining objective and method for NoiseVAE are well introduced with details.

### Significance
1. Exploring the conformational space efficiently is of significance in quantum chemical simulations. NoiseVAE provides a way to efficiently sample conformations around a given geometry.

**Weaknesses:**

While I appreciate the amount of information and results presented in the paper, I find the experiments and results a bit difficult to follow.

- The authors showed proof that the proposed DenoiseVAE provides a higher theoretical evidence lower bound (ELBO) guarantee for the real conformation distribution of isoenergetic molecules.

- Subsequently, to show the effectiveness of the DenoiseVAE, the authors conducted downstream tasks with the pre-trained DenoiseVAE for predicting molecular properties.

- However, the authors did not provide a clear explanation of how DenoiseVAE was used in the downstream molecular property prediction tasks. To the best of my understanding, the DenoiseVAE was used to generate noise distributions for the input molecules. However, how were these noise distributions leveraged was not clear to me.

- The tables with results are also a bit misleading. For example, tables 1 and 2 mentioned "force prediction", but the 12 properties in Table 1 were not force-related and the values in Table 2 seemed more energy-related than force-related.

- In addition, the paper lacks details on the training process of the NoiseVAE model. Specifically, the pretraining is performed with the PCQM4Mv2 dataset, but the paper only briefly mentions that the dataset has 3.4 million organic molecules.

**Questions:**

The authors are encouraged to provide more details concerning the above-mentioned weaknesses. For example,
1. What is the significance of the PCQM4Mv2 dataset in the pretraining of the NoiseVAE model?
   - How does the dataset help in learning atom-specific noise distributions?
1. How are the noise distributions generated by DenoiseVAE used in the downstream molecular property prediction tasks?
   - What is the architecture of the property predictor(s) used in the downstream tasks? What type of model is used?
   - How are the noise distributions used in the property prediction tasks, in addition to the input 3D molecular geometry?
1. What are the correct metrics/criteria for the results in Tables 1 and 2?

Additionally, the authors may consider addressing the following questions:
1. The paper provides proof that the DenoiseVAE provides a higher theoretical evidence lower bound (ELBO) guarantee for the real conformation distribution of isoenergetic molecules. Is there a more direct way to show the correctness of the noise distributions generated by DenoiseVAE with the ground truth by first-principles calculations?
   - For example, in Figure 4, the authors showed the generated noise distributions for two molecules. How close are these distributions when benchmarked against first-principles simulations?
1. How does the DenoiseVAE model help with conformational sampling in quantum chemical simulations?
   - The DenoiseVAE model generates noise distributions that are specific to a 3D molecular geometry (to the best of my understanding). By sampling from these noise distributions, are the sampled conformations more likely to be energetically favorable/stable than randomly sampling from a preset Gaussian distribution?

---

> ### Author Response · Authors · 2024-11-20
>
> We sincerely appreciate the reviewers for their thorough evaluation of our work and their positive recognition of its quality. We also highly value the insightful questions and suggestions raised by the reviewers. We have carefully addressed each of these issues and incorporated the relevant content into the newly uploaded PDF.
>
> ***Question 1:***
> > The paper lacks details on the training process of the NoiseVAE model. What is the significance of the PCQM4Mv2 dataset in the pretraining of the NoiseVAE model? How does the dataset help in learning atom-specific noise distributions?
>
> Thank you for your valuable feedback and **we have added a more detailed introduction to all the dataset in the Appendix A.7, A.8, A.9, A.10 of the newly uploaded pdf**. In essence, our method learns the molecular force field surrounding the equilibrium conformation through the pretraining process, as detailed in Section 3.1 of the paper. By explicitly considering the differences in the force fields acting on each atom within the molecule, our method enables more accurate learning. For this learning process, accurate coordinate data and sufficient data volume are important guarantees. The PCQM4Mv2 dataset perfectly satisfies these requirements, offering 3.4 million molecular data points and DFT-computed accurate equilibrium molecular conformation data. Therefore, consistent with previous studies[1][2][3], we select the PCQM4Mv2 dataset for pretraining. **In addition, we also conduct experiments on another pre-training dataset and also achieve good performance in downstream tasks. Please refer to Table 13 in the Appendix of the paper**.
>
> **PCQM4Mv2 dataset and the detailed pre-training process**:
> The dataset used during pretraining is PCQM4Mv2, a large-scale chemical dataset based on molecular graph structures which is widely used for training and evaluating molecular representation learning models. It contains 3.4 million organic molecules, each associated with a set of three-dimensional coordinates for all atoms within the molecule and the energy value of the molecular conformation at these coordinates. This energy value is obtained through conformational optimization using the Density Functional Theory (DFT) method, ensuring that the corresponding molecular conformation represents the equilibrium conformation (the lowest-energy stable state). Our pretraining phase aims to learn the true distribution of molecular conformations \( p(x) \), as described in Section 3.1 of our paper. Since the true force fields required for learning p(x)  are unavailable, we approximate the force field learning through a noise-based denoising pretraining approach on the PCQM4Mv2 dataset. This approximation is supported by prior research[4], which has demonstrated the equivalence between denoising learning and force field learning. Specifically, we input the stable conformations of each molecule into the Noise Generator to obtain the noise distribution to be added to each atom. After sampling and adding noise, the noisy molecular conformations are fed into the Denoising Module to predict the added noise. This denoising learning process is equivalent to molecular force field learning.
>
> [1]Zaidi, Sheheryar, et al. "Pre-training via Denoising for Molecular Property Prediction." The Eleventh International Conference on Learning Representations.
>
> [2]Feng, Shikun, et al. "Fractional denoising for 3d molecular pre-training." International Conference on Machine Learning. PMLR, 2023.
>
> [3]Ni, Yuyan, et al. "Sliced Denoising: A Physics-Informed Molecular Pre-Training Method." The Twelfth International Conference on Learning Representations.
>
> [4] Vincent, Pascal. "A connection between score matching and denoising autoencoders." Neural computation 23.7 (2011): 1661-1674.

---

> ### Author Response · Authors · 2024-11-20
>
> ***Question 2:***
> > How are the noise distributions generated by DenoiseVAE used in the downstream molecular property prediction tasks? What is the architecture of the property predictor(s) used in the downstream tasks? What type of model is used? How are the noise distributions used in the property prediction tasks, in addition to the input 3D molecular geometry?
>
> We are sorry for the confusion and we would like to clarify that **our learned noise distribution is not directly utilized in downstream tasks; instead, the molecular representations obtained during pretraining are applied to these tasks**.
>
> Specifically, our method adopts a **pretraining-finetuning paradigm, comprising two stages: unsupervised pretraining and supervised downstream fine-tuning**. In the first stage, a large amount of unlabeled data is used for denoising pretraining to obtain high-quality molecular representations. These representations are subsequently fine-tuned on labeled downstream datasets to generate predictions for specific tasks. The noise-based denoising learning process is conducted exclusively during the first stage, i.e., the self-supervised pretraining phase. **During the downstream fine-tuning phase, the Noise Generator is discarded, and only the pre-trained Denoising Module is employed for supervised fine-tuning.** The Denoising Module is a 7-layer equivariant graph neural network (EGNN) with a hidden dimension of 128, as detailed in Appendix A.15 of this paper. In the fine-tuning phase, molecular data from the downstream dataset is input into the Denoising Module to derive molecular representations, which are then fed into a 2-layer perceptron (MLP) prediction head to produce predictions of molecular properties. This entire fine-tuning process is conducted under a supervised learning framework. Finally, the fine-tuned Denoising Module+MLP model is evaluated on downstream test sets, yielding the property prediction results reported in the tables.
>
> ***Question 3:***
> > What are the correct metrics/criteria for the results in Tables 1 and 2?  Tables 1 and 2 mentioned "force prediction", but the 12 properties in Table 1 were not force-related and the values in Table 2 seemed more energy-related than force-related.
>
> Thank you for your helpful comments and **we have added detailed introductions to the two datasets in Appendix A.8 and A.9**. We are sorry that we identify a typo in the title of Table 1, which should be corrected to "Performance on QM9 property prediction." This has been rectified in the newly uploaded PDF.
>
> **The tasks on the QM9 dataset involve predicting static molecular properties based on quantum chemical calculations (regression tasks).** The evaluation metric is the mean absolute error (MAE) between the predicted and true values, and the corresponding units for each property are detailed in Table 1.
>
> **The MD17 dataset**, on the other hand, comprises time-series data obtained through molecular dynamics simulations for each molecule. **The task involves molecular force field prediction (also a regression task)**, and the evaluation metric is also the mean absolute error (MAE) between the predicted and true values, with units expressed as $kcal$ $mol^{-1} Å^{-1} $.
>
> ***Question 4:***
> > The paper provides proof that the DenoiseVAE provides a higher theoretical evidence lower bound (ELBO) guarantee for the real conformation distribution of isoenergetic molecules. Is there a more direct way to show the correctness of the noise distributions generated by DenoiseVAE with the ground truth by first-principles calculations? For example, in Figure 4, the authors showed the generated noise distributions for two molecules. How close are these distributions when benchmarked against first-principles simulations?
>
> **Nice suggestion! We have conducted additional experiments and visualizations to provide a more intuitive explanation of this issue, which has been included in Figure 5 in Section 5.4 of the newly uploaded PDF**. Due to the difficulty in obtaining precise molecular force fields through first-principles calculations using DFT, we calculate the energy changes before and after noise perturbations for two molecules shown in Figure 5 using RDKit library functions under an empirical force field.
>
> In these experiments, we apply noise perturbations of the same scale to one atom in the molecule at a time and compute the energy changes between the perturbed and original conformations. The experimental results align well with relevant
> chemical constraints. For atoms that are more sensitive to noise perturbations (i.e., perturbations that significantly impact the molecule’s overall energy), we apply smaller-scale noise. Conversely, for atoms that are less sensitive, we apply larger-scale noise.

---

> ### Author Response · Authors · 2024-11-20
>
> ***Question 5:***
> > How does the DenoiseVAE model help with conformational sampling in quantum chemical simulations? The DenoiseVAE model generates noise distributions that are specific to a 3D molecular geometry (to the best of my understanding). By sampling from these noise distributions, are the sampled conformations more likely to be energetically favorable/stable than randomly sampling from a preset Gaussian distribution?
>
> **Your understanding is absolutely correct**. Our method indeed achieves adaptive learning of atom-specific noise distributions, **enabling the sampled molecular conformations to exhibit lower energy values and greater stability**, which are more conducive to molecular representation learning. It is worth emphasizing that the primary objective of our method is to achieve better molecular representation learning, with improved conformation sampling serving as a technique to facilitate better representation learning. **As shown in Figure 3 of the paper, we calculate the energy changes of molecular conformations before and after noise perturbations under different noise addition schemes**.
>
> Compared to fixed-scale noise with the same prior distribution, the energy change in molecular conformations perturbed by our model is only **10.36%** relative to the stable conformations before noise addition, which is significantly smaller than the **96.84%** observed under fixed-scale noise. Furthermore, the energy change produced by our method is even substantially lower than that of fixed-scale noise with a variance of 0.001, which results in an energy change of **32.17%**. These results intuitively demonstrate that the molecular conformations sampled by our method possess higher stability.

---

> ### Author Response · Authors · 2024-11-24
>
> Dear Reviewer,
>
> Thank you once again for your professional and insightful feedback. We greatly value your suggestion to visualize noise sampling, which we believe will enhance the clarity and intuitiveness of our case study. We have carefully addressed your specific points on the model design and experimental setup, and we hope these adequately address all your concerns.
>
> As the discussion period is nearing its end, we would greatly appreciate it if you could let us know whether our responses and the revised manuscript sufficiently resolve your questions or if you have any additional comments.
>
> Best regards,
>
> The Authors

---

> > ### Comment · Reviewer_D6p5 · 2024-11-26
> >
> > Thank the authors for the discussions.
> >
> > I do agree with the other reviewers that comparing the results with more physically related methods will make the claims more convincing, such as DFT, semi-empirical methods, force field-based methods, or even machine learning potential-based methods.
> >
> > For example,  in Figure 5, the authors showed the learned noise distribution for each of the atoms. The results look reasonable, but this is not difficult to predict with general organic chemistry domain knowledge. For example, triple bonds and three-member rings are supposed to be rigid and thus have low noise distribution. In addition, hydrogens seem to be excluded from the visualization, which is unreasonable as they are important, although they are typically implicitly notated. What would be more significant is to show that the learned distribution is **quantitatively** more accurate against DFT, semi-empirical methods, or force field-based methods.
> >
> > The improvements in Table 1 do not appear significant to me. Although the baselines may be already very robust, I am leaning to think that learned representation did not help much since the authors have not shown the abovementioned evidence.
> >
> > I will keep my original score.

---

> > > ### Author Response · Authors · 2024-12-02
> > >
> > > We would like to express our sincere gratitude for your previous response! In light of the concerns you raised, we have conducted additional experiments and provided further clarifications on the key aspects of our method. We hope that these supplementary details and updates thoroughly address your concerns.
> > >
> > > As the discussion phase is nearing its end, we would greatly appreciate your feedback and would be truly grateful for any additional insights or comments you may have. Thank you once again for your time and consideration. We kindly look forward to your reply.
> > >
> > > Best regards,
> > >
> > > The Authors

---

> ### Author Response · Authors · 2024-11-27
>
> **We sincerely appreciate your time and thoughtful feedback**. However, we would like to clarify an important aspect of our work. The primary goal of our study is not to precisely conduct exact dynamics simulations, or generate and compute highly accurate molecular conformations. Instead, **our focus is to leverage the process of approximating molecular force field to develop a better molecular representation learning method**.
>
> Our choice to pursue a self-supervised molecular representation learning method stems from the challenges posed by Density Functional Theory (DFT) calculations. **While DFT offers high accuracy, it is prohibitively expensive in terms of both time and computational cost. Our work aims to address this limitation by enabling more efficient unsupervised representation learning using a large amount of easily accessible unlabeled data, thereby narrowing the performance gap with DFT calculations without requiring additional DFT computations. This approach enhances generalization to broader application scenarios.** Similar to our approach, **SliDe [1]** also embraces the idea of improving molecular representations by designing noise in a more physically meaningful way, which achieves excellent performance in downstream tasks.
>
> **Based on your suggestions, we also conduct experiments using DFT-based energy labels**. Since our large-scale pre-training dataset relies on DFT to compute energy and extract the most stable conformers with the lowest energies, we perform supervised pretraining using the true energy labels derived from DFT calculations and compare it with our method. **Due to our limited computational resources, we have only recently obtained the experimental results, which are summarized in the table below**.
>
> | |$\mu$ |$\alpha$ |$\epsilon_{HOMO}$| $\epsilon_{LUMO}$ |$\Delta\epsilon$ | $< R^{2} >$ | ZPVE | $U_{0}$ | $U$ | $H$ | $G$ | $C_{v}$|
> |--|--|--|--|--|--|--|--|--|--|--|--|--|
> |SliDe (force field-based) |0.0087|0.0366|13.6|12.3|26.2|0.3405|1.521|4.28|4.29|4.26|5.37|0.019|
> |Ours (DFT-labeled) |0.017|0.0663|24.74|19.6|43.27|0.136|1.515|8.74|8.66|8.61|8.93|0.026|
> |Ours |**0.0079**|0.0650|14.2|**11.9**|**26.0**|**0.062**|**1.028**|4.31|**4.03**|**4.19**|**5.35**|**0.015**|
>
> The results demonstrate that our method significantly outperforms both the DFT-labeled and force field-based methods. **Notably, similar trends have been observed in previous studies**, such as GeoSSL-DDM [2], where the unsupervised pretraining method based on denoising consistently outperforms DFT-labeled approach across all downstream tasks.
>
> **Regarding the results presented in Table 1, our method achieves state-of-the-art performance among all self-supervised learning methods (to the best of our knowledge). It is worth mentioning that the prediction tasks on the QM9 dataset are highly challenging, and recent molecular representation methods have struggled to achieve substantial improvements**. For example, **SliDe [1]** improves upon **Frad [3]** by introducing additional physics constraints, but even for the task with the largest performance gain among the 12 tasks, the improvement is only **23.6%**. **In contrast, our method achieves a performance gain of up to 81.8% compared to SliDe for the task with the largest improvement , which is a remarkable result**. This strongly demonstrates the potential and contribution of our method to advancing molecular representation learning in this domain.
>
> **Thank you again for your patient response and we sincerely hope our response could address your concerns**.
>
> [1] Ni, Yuyan, et al. "Sliced Denoising: A Physics-Informed Molecular Pre-Training Method." The Twelfth International Conference on Learning Representations.
>
> [2] Liu, Shengchao, Hongyu Guo, and Jian Tang. "Molecular Geometry Pretraining with SE (3)-Invariant Denoising Distance Matching." The Eleventh International Conference on Learning Representations.
>
> [3] Feng, Shikun, et al. "Fractional denoising for 3d molecular pre-training." International Conference on Machine Learning. PMLR, 2023.

---

> > ### Comment · Reviewer_D6p5 · 2024-12-03
> >
> > Thanks for the additional results.
> >
> > I still believe that a more ground-truth comparison of the generated noise to DFT or force-field simulations is necessary for the application of the method. The additional results did not address this concern. My concern was more related to the generated noise instead of using DFT for downstream prediction. In other words, how is the generated noise compared to the actual force field-based simulations? Are the noise levels quantitatively accurate?
> >
> > I do acknowledge that the results show considerable improvements. I will thus raise my score to 6.

---

> > > ### Author Response · Authors · 2024-12-03
> > >
> > > **Thank you very much for your recognition of our work! Your valuable suggestions have significantly enhanced the overall quality of our research, which is a great honor for us.**
> > >
> > > Regarding the issue of the distance between noise and the true distribution, we address this concern through theoretical proofs and two experimental studies in our paper. **In Section 4, we provide theoretical proof that our method achieves a higher evidence lower bound (ELBO) compared to fixed-scale noise methods. This implies that the molecular conformation distribution sampled under our noise-adding scheme is theoretically closer to the true distribution**. Furthermore, **in Figure 3** of our paper, we visualize the energy variations of molecular conformations under different noise perturbation patterns. The results demonstrate that **the conformations sampled by our method exhibit significantly lower energy (indicating greater stability) compared to those generated with fixed-scale noise perturbations**. These findings support our motivation to focus sampling around stable conformations. Additionally, **in Figure 5**, we provide a visualization of the noise learned by the model for specific examples. The results reveal an inverse relationship between the noise added to individual atoms and the energy sensitivity of those atoms. This observation suggests that **our model captures the physical properties of each atom to a certain extent**.
> > >
> > > **Of course, we acknowledge that the most direct approach to demonstrating the distance between our noise and the true distribution would be to compare the conformation distribution obtained using real DFT calculations with our method's distribution.** However, due to the difficulty of acquiring a sufficient number of DFT data points enough to calculate the true distribution within a limited timeframe, we employ the aforementioned approaches for preliminary exploration. We plan to continue this line of investigation in the future, including integrating real DFT data with our method.
> > >
> > > **Once again, thank you for your thorough review and the valuable suggestions you provided during the discussion. They are invaluable in guiding the future direction of our work!**

---

### Meta-Review · Area_Chair_5EM5 · 2024-12-17

**Metareview:**

This paper introduces DenoiseVAE, a denoising-based 3D molecular pre-training framework that learns molecule-adaptive, atom-specific noise distributions. Unlike prior methods relying on uniform or hand-crafted noise, the proposed approach employs a Noise Generator to produce customized noise distributions for each atom, while using a standard denoising objective to approximate molecular force fields. This leads to potentially more physically informed molecular representations.

All reviewers acknowledged the novelty and reasonableness of learning atom-specific noise distributions, and they found the empirical improvements over baseline methods to be well demonstrated. However, several reviewers raised concerns about the physical fidelity of the generated conformations. They suggested more direct comparisons with ground-truth force fields or DFT-calculated distributions to better validate the physical authenticity. While the authors did provide some responses and additional experiments, these concerns were not fully resolved during the rebuttal. Additionally, some reviewers recommended integrating known rotational, vibrational, or quantum constraints into the model or loss function. Although the authors partially addressed these points, the reservations were not entirely alleviated.

Overall, all reviewers reached a consensus that the paper presents a meaningful contribution to the denoising paradigm for molecular representation learning, despite reservations regarding its physical fidelity. The AC agrees with the reviewers and recommends acceptance.

**Additional Comments On Reviewer Discussion:**

During the rebuttal and discussion phase, reviewers requested clarifications regarding how noise distributions are used downstream, how datasets and metrics are defined, and how the model’s physical fidelity could be strengthened through direct comparisons to ground-truth force fields (D6p5, Ho1Z, NLii, R3gr). The authors addressed some points by revising their manuscript and adding details, such as clarifying their training protocols, explaining dataset usage, and visualizing loss curves and noise distributions (D6p5, Ho1Z, NLii). They also explored a variant incorporating rotational modeling and provided additional experiments and theoretical reasoning (R3gr). While these efforts resolved several concerns—leading some reviewers to raise their scores—questions about the exact physical authenticity remained (D6p5, R3gr). However, the majority of reviewers recognized the authors’ responsiveness and the paper’s improvements, ultimately supporting acceptance (Ho1Z, NLii).

---

### Decision · Program_Chairs · 2025-01-22

Accept (Poster)